# Memory as Dynamics: Learning Reliability-Guided Predictive Models for Online Video Perception

**Minwoo Kim** [1]   **Sang Min Yoon** [1]

## Abstract

Predictive memory has recently emerged as a powerful mechanism for online video models, enabling temporal reasoning beyond static memory banks. However, we observe a paradoxical phenomenon in which predictive memory often exacerbates drift under occlusion or rapid motion, as inaccurate predictions contaminate the internal state and lead to irreversible identity loss. We identify this failure as a reliability mismatch: predictive dynamics are applied uniformly despite high uncertainty and weak observations. To address this issue, we reinterpret video memory as a dynamic latent process rather than a static buffer. Building on this insight, we introduce Reliability-Guided Predictive Memory (RPM), a framework that explicitly regulates when and how predictive dynamics should influence online video perception. RPM integrates a latent world model based on state-space dynamics to generate predictive priors, while employing a reliability-aware fusion policy that suppresses unreliable predictions during occlusion. We instantiate RPM on a SAM2-based model and evaluate it on visual object tracking benchmarks. Experiments demonstrate that our method reduces drift after occlusion, consistently outperforming strong baselines that rely on either static memory or unconditional predictive modeling. We further show that this reliability-guided mechanism transfers to video object segmentation across multiple standard benchmarks. These findings establish that predictive memory is beneficial only when its reliability is explicitly modeled, and define a general principle for robust online video perception. The code is available at https://github.com/minwookim01/memory-as-dynamics.

[1]HCI Lab, College of Computer Science, Kookmin University, Seoul, Korea. Correspondence to: Sang Min Yoon <smyoon@kookmin.ac.kr>.

*Proceedings of the 43$^{rd}$ International Conference on Machine Learning*, Seoul, South Korea. PMLR 306, 2026. Copyright 2026 by the author(s).

## 1. Introduction

Memory is a fundamental component of online video perception, providing the temporal continuity required to connect isolated frames into a coherent visual stream. Modern video foundation models, most notably the Segment Anything Model 2 (SAM2) (Ravi et al., 2025; Kirillov et al., 2023), implement this capability through internal states that accumulate and attend to information from past observations. While static memory mechanisms have shown strong performance in propagating high-fidelity appearance over short temporal windows, they are intrinsically limited when visual evidence is weak or missing, such as during prolonged occlusion, abrupt motion, or severe distractors. In these cases, the absence of explicit temporal dynamics makes relying solely on past observations insufficient, motivating the use of predictive mechanisms that can extrapolate future states. Predictive memory addresses this limitation by modeling latent dynamics and generating future states beyond what is directly observed. Recent advances in world models (Hafner et al., 2019; 2025) demonstrate the potential of such predictive abstractions for temporal reasoning beyond direct perception.

However, predictive memory is not universally beneficial when integrated without regulation. In practice, we observe a counterintuitive failure phenomenon when predictive priors are unconditionally injected into memory under uncertainty. In challenging video sequences involving long occlusion or ambiguous re-acquisition, speculative predictions may be committed to the memory state while observation quality is low. This leads to memory contamination, where hallucinated features accumulate over time and overwhelm reliable historical information, ultimately causing irreversible identity drift after occlusion. As we demonstrate empirically on standard tracking benchmarks (Fan et al., 2019; 2021; Huang et al., 2021), such unregulated predictive memory can underperform simpler static-memory baselines in precisely these failure regimes. This observation reveals a fundamental gap between predictive capability and practical robustness in online video perception. We argue that this failure arises from a reliability mismatch in how predictive memory is integrated. Existing approaches often treat predicted latent states as equally reliable as observation-driven

ones, even under severe uncertainty. During occlusion or rapid motion, committing a single speculative prediction to memory amplifies error accumulation. Our goal is therefore not to model multi-modal futures, but to prevent unreliable predictions from being permanently written into memory.

Motivated by this insight, we reinterpret video memory as a dynamic latent process whose evolution must be explicitly governed by reliability. We propose Reliability-Guided Predictive Memory (RPM), a general framework that augments foundation video models (Ravi et al., 2025) with state-space latent dynamics (Gu et al., 2022; Gu & Dao, 2024; Dao & Gu, 2024) while adaptively controlling their influence. Unlike prior reliability heuristics that primarily gate updates based on observation confidence alone, RPM explicitly separates short-term perceptual fusion from long-term memory accumulation by regulating the perceptual influence of predictive dynamics using a reliability signal that combines prediction-observation consistency, observation-driven confidence, and observation-driven spatial reference. This design prevents speculative predictive features from contaminating long-term memory during periods of uncertainty. We apply RPM to a SAM2-based foundation video model and evaluate it on visual object tracking, a representative online video perception setting (Kristan et al., 2020; Čehovin Zajc, 2020) where memory reliability is paramount and post-occlusion drift is a dominant failure mode. Recent SAM2-based tracking (Yang et al., 2024; Videnovic et al., 2025; Chen et al., 2025a) approaches highlight both the promise and the limitations of memory-centric designs in this setting. Experimental results show that RPM consistently reduces post-occlusion drift and improves robustness relative to strong static-memory models and methods that employ unconditional predictive dynamics. Beyond empirical gains, our findings support a general principle for online video perception: predictive memory is beneficial only when its reliability is explicitly modeled. This principle provides a robust foundation for integrating world-model-based reasoning into video foundation architectures without sacrificing long-term stability. The main contributions of this work are as follows:

- We identify a reliability mismatch in online video perception, where unregulated commitment of predictive memory under uncertainty contaminates internal states and leads to irreversible identity drift.

- We propose RPM, a framework that treats memory as a dynamic latent process and modulates the perceptual influence of predictive memory via a reliability-aware control signal.

- We demonstrate on visual object tracking benchmarks that reliability-guided predictive memory substantially reduces post-occlusion drift and improves robustness over strong static-memory and unconditional predictive baselines.

## 2. Related Work

### 2.1. Memory in Online Video Models

Memory is a fundamental mechanism in online video perception, enabling models to propagate information across time under appearance changes and partial observability. Many tracking and video segmentation systems rely on memory banks that store features from previous frames and retrieve them through attention or matching. This paradigm has been formalized in recent foundation video models such as SAM2 (Ravi et al., 2025), which maintain explicit memory representations to accumulate visual evidence over time. Related memory-centric designs have also shown strong performance in video object segmentation by emphasizing feature reuse across frames (Oh et al., 2019). Despite their effectiveness, static memory mechanisms remain intrinsically dependent on reliable observations. When visual evidence is weak or missing, such as during prolonged occlusion or abrupt motion, memory updates become ambiguous and error-prone. In these regimes, memory banks may propagate outdated or incorrect information, leading to drift. Importantly, most memory-based approaches implicitly assume that past observations remain trustworthy, and do not explicitly account for the reliability of stored or retrieved memory under uncertainty.

### 2.2. Predictive Memory and Latent Dynamics

To overcome the limitations of purely retrospective memory, predictive modeling has emerged as a complementary paradigm. World models (Hafner et al., 2025) aim to capture latent dynamics that explain how observations evolve over time, enabling long-horizon reasoning beyond direct perception. Recent work in predictive representation learning (Bardes et al., 2024) suggests that anticipating future latent states can lead to strong and transferable visual representations without relying on explicit pixel-level reconstruction. These advances suggest that predictive memory can serve as a compact abstraction of temporal structure.

Recent progress in state-space models (Gu et al., 2022; Gu & Dao, 2024; Dao & Gu, 2024) provides efficient tools for modeling long-range dependencies through latent state evolution. Structured State Space models and their variants enable linear-time sequence modeling while maintaining expressive dynamics. While these models excel at capturing temporal structure, they are typically evaluated as standalone sequence learners or offline predictors. When predictive dynamics are integrated into online video perception systems, predicted states are often injected into memory without explicit regulation. As a result, prediction errors may accumulate and dominate the internal state, particularly under occlusion or multi-modal futures. Existing predictive approaches rarely address how uncertainty in latent dynamics should modulate memory integration, leaving online systems vulnerable to drift.

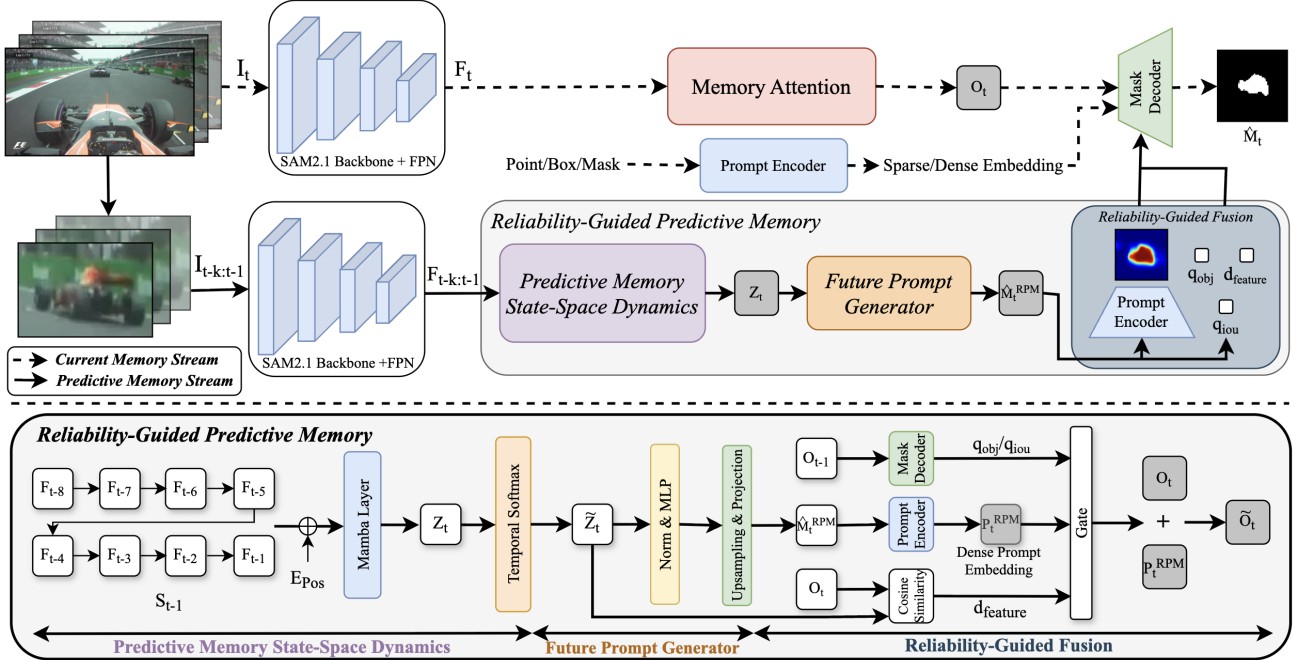

*Figure 1.* Overview of Reliability-Guided Predictive Memory for online video perception. A predictive prior is generated from historical memory via latent state-space dynamics. Its influence is regulated by a reliability signal and injected as a prompt for perceptual fusion, while preventing unreliable predictive information from being committed to long-term memory. This separation prevents memory contamination and enables robust recovery under occlusion.

## 2.3. Reliability in Memory-Based Video Models

Reliability and uncertainty have long been central concerns in tracking and state estimation. Classical filtering (Kalman, 1960) approaches regulate state updates by weighting observations according to confidence and noise statistics. In modern tracking systems, reliability-aware mechanisms (Lukežič et al., 2017; Kristan et al., 2018; Park et al., 2025) have been explored to suppress unreliable observations or distractors, such as channel and spatial reliability weighting or selective memory updates. However, these methods are primarily designed to regulate observation-driven updates and do not directly address how learned, high-dimensional predictive dynamics should be selectively activated or suppressed when integrated into online perception models.

Recent SAM2-based tracking (Yang et al., 2024; Videnovic et al., 2025; Chen et al., 2025a) approaches highlight both the promise and the limitations of memory-centric designs. Methods such as SAMURAI (Yang et al., 2024) incorporate motion-aware memory updates, while distractor-aware memory schemes, such as DAM4SAM (Videnovic et al., 2025), aim to reduce interference from similar objects. HiM2SAM (Chen et al., 2025a) further enhances SAM2 with hierarchical motion modeling and memory optimization for long-term tracking. Although these methods improve memory management, they do not model predictive dynamics, and thus remain inherently observation-driven,

unable to exploit reliability signals when observations degrade under occlusion. In contrast, our work treats reliability as a first-class principle that governs how predictive memory influences state evolution under occlusion, preventing memory contamination at its source.

## 3. Reliability-Guided Predictive Memory Framework

### 3.1. Problem Formulation

We formulate online video perception as a sequential latent state estimation problem under partial observability. At each time step $t$, the system maintains an internal latent state $\mathbf{s}_t$ that summarizes temporal information accumulated from past observations. Given the current observation $\mathbf{o}_t$, extracted from the input frame, the model updates its internal state to support downstream prediction and segmentation.

In memory-based video models, such as SAM2, this process is commonly expressed as:

$$\mathbf{s}_t = \mathcal{U}(\mathbf{s}_{t-1}, \mathbf{o}_t), \tag{1}$$

where $\mathcal{U}(\cdot)$ denotes a state update function. In practice, $\mathbf{s}_t$ is implemented as a memory bank that stores and retrieves features from previous frames, and $\mathcal{U}$ relies primarily on observation-driven updates through memory attention mechanisms.

Under occlusion or ambiguous observations, however, $\mathbf{o}_t$

provides weak or unreliable evidence about the true underlying state. To compensate for missing information, predictive memory introduces an additional latent variable $\mathbf{z}_t$ that models the temporal evolution of the state. In this work, we model predictive dynamics using a deterministic state transition:

$$\mathbf{z}_t = \mathcal{P}(\mathbf{s}_{t-1}), \tag{2}$$

where $\mathcal{P}$ extrapolates the current state from historical context.

A common practice in existing predictive approaches is to integrate $\mathbf{z}_t$ as an unconditional component in the state update, effectively treating predictive dynamics and observations as equally reliable. This leads to a fused update of the form:

$$\mathbf{s}_t = \mathcal{F}(\mathbf{s}_{t-1}, \mathbf{o}_t, \mathbf{z}_t), \tag{3}$$

where noisy predictive estimates are injected into memory without a selective bottleneck. When the predictive prior is inaccurate, errors accumulate in the internal state and cause irreversible memory contamination after occlusion.

In contrast, we argue that predictive memory should be treated as a conditional prior whose influence is governed by its reliability. To make the resulting system flow concrete, we briefly summarize how RPM augments this baseline. RPM introduces a predictive latent $\tilde{\mathbf{z}}_t$ via state-space dynamics over the recent memory tracklet, and a reliability signal $\rho_t$ derived from observation confidence and prediction-observation consistency. As shown in Fig. 1, $\tilde{\mathbf{z}}_t$ is then injected into the decoder as a prompt only when $\rho_t = 1$, leaving the underlying memory update $\mathcal{U}$ observation-driven. This formulation forms the basis of our RPM framework, detailed in the following sections.

### 3.2. Predictive Memory via State-Space Dynamics

We model predictive memory using state-space dynamics to extrapolate the internal memory state when observation-driven updates become unreliable. To this end, we adopt deterministic state-space dynamics that generate a stable predictive prior for online inference.

Let $\mathbf{F}_{t-k} \in \mathbb{R}^{N \times D}$ denote the observation-driven memory features extracted from frame $t - k$. A temporal tracklet is formed from the past $K$ frames $\{\mathbf{F}_{t-K}, \ldots, \mathbf{F}_{t-1}\}$, which are concatenated as memory tokens to form the tracklet memory $\mathbf{s}_{t-1} \in \mathbb{R}^{(K \cdot N) \times D}$. A fixed sinusoidal temporal positional embedding $\mathbf{E}_{\text{pos}}$ is added to preserve temporal order.

We introduce an internal hidden state sequence $\{h_{t,k}\}_{k=1}^{K}$ governed by a discretized state-space model:

$$h_{t,k} = \bar{\mathbf{A}} h_{t,k-1} + \bar{\mathbf{B}} \operatorname{Pool}(\mathbf{s}_{t-k}), \tag{4}$$

where $\mathbf{s}_{t-k}$ is the observation-driven memory from the $k$-th most recent frame and $\operatorname{Pool}(\cdot)$ aggregates token features.

The predictive latent sequence is obtained as

$$\mathbf{z}_{t,k} = \mathbf{C} h_{t,k}, \quad k = 1, \ldots, K, \tag{5}$$

where $\mathbf{C}$ projects the internal dynamics state to the predictive memory space.

In this formulation, $\bar{\mathbf{A}}$ propagates the hidden state over time, $\bar{\mathbf{B}}$ integrates new evidence from the tracklet memory, and $\mathbf{C}$ projects the hidden state onto the predictive latent space. We provide the concrete instantiation of these operators in Table 8.

The state-space dynamics operate over a short temporal tracklet within a single time step, yielding a sequence of predictive latent features $\{\mathbf{z}_{t,k}\}_{k=1}^{K}$ aligned with the positions of the input window. To obtain a single stable predictive memory prior for downstream processing, we aggregate this sequence using softmax-weighted temporal pooling:

$$\alpha_k = \frac{\exp(g(\mathbf{z}_{t,k}))}{\sum_{j=1}^{K} \exp(g(\mathbf{z}_{t,j}))}, \qquad \tilde{\mathbf{z}}_t = \sum_{k=1}^{K} \alpha_k \, \mathbf{z}_{t,k}, \tag{6}$$

where $g(\cdot)$ denotes a lightweight scoring function. The aggregated latent $\tilde{\mathbf{z}}_t$ serves as the final predictive memory prior for subsequent processing.

### 3.3. Future Prompt Generator

The predictive memory prior $\tilde{\mathbf{z}}_t$ encodes anticipated temporal evolution but is not directly compatible with the SAM2 mask decoder. We therefore transform it using a lightweight module consisting of normalization and a two-layer MLP, followed by spatial projection and upsampling, to obtain a decoder-aligned dense prompt $\hat{\mathbf{M}}_t^{\text{RPM}}$ for mask decoding. In contrast, the original predictive latent $\tilde{\mathbf{z}}_t$ is preserved and directly used for reliability evaluation without additional transformation.

### 3.4. Reliability Signal Modeling

Predictive memory offers valuable guidance when observation-driven evidence is weak, yet its reliability varies over time, especially under occlusion where prediction–observation discrepancies become ambiguous. To address this challenge, we introduce a reliability-aware mechanism that decomposes reliability into complementary signals to regulate predictive activation and evaluate the trustworthiness of observation-driven spatial references during state evolution.

To determine whether predictive memory can be safely *activated*, we measure a prediction-observation consistency between observation-driven memory features $\mathbf{o}_t$ and predictive features $\tilde{\mathbf{z}}_t$:

$$d_t^{\text{feature}} = 1 - \frac{\langle \mathbf{o}_t, \tilde{\mathbf{z}}_t \rangle}{\|\mathbf{o}_t\| \, \|\tilde{\mathbf{z}}_t\|}, \tag{7}$$

where $\mathbf{o}_t$ and $\tilde{\mathbf{z}}_t$ share the same feature dimensionality. Because $\tilde{\mathbf{z}}_t$ is derived from past memory tokens whereas $\mathbf{o}_t$ is extracted from the current frame, comparing them without spatial alignment would conflate prediction error with target displacement. To disentangle the two, we align both features in a shared object-centric frame through target-centered cropping: a single crop window, centered on the most recently reliable target location and governed by a common scale factor $\alpha_{\mathrm{crop}}$, is applied identically to $\mathbf{o}_t$ and $\tilde{\mathbf{z}}_t$. The resulting consistency signal captures the agreement between prediction and observation and serves as the primary criterion for predictive memory activation.

In addition, we extract decoder-derived confidence signals from the underlying SAM2-based video model to assess the reliability of recent observations:

$$\left(q_t^{\mathrm{obj}}, q_t^{\mathrm{iou}}\right) = (\lambda_o, \lambda_i) \odot \mathcal{D}(\mathbf{o}_{t-1}), \tag{8}$$

where $\mathcal{D}(\cdot)$ denotes the decoder head of the SAM2-based model, which outputs the objectness logit and the sigmoid-normalized predicted mask IoU associated with the observation-driven prediction from the previous frame. While $q_t^{\mathrm{iou}} \in [0, 1]$ represents an observation-driven confidence that reflects the reliability of recent tracking outputs and informs predictive activation, $q_t^{\mathrm{obj}}$ serves as an observation-driven spatial reference that assesses the reliability of using the current observation as a geometric anchor for spatial grounding.

### 3.5. Reliability-Guided Fusion

We describe how predictive memory is regulated during inference. Rather than indiscriminately injecting predictive outputs, we control predictive memory along two complementary aspects: when predictive reasoning is activated and how predictive cues are spatially anchored.

**Reliability-Guided Predictive Activation.** Unlike observation-driven cues, predictive prompts introduce model-driven biases that can persist and accumulate once they are injected into the decoding process, particularly under uncertain conditions. Consequently, we avoid continuous fusion of predictive and observation features, as soft or learnable blending mechanisms may still propagate partially unreliable predictions and lead to progressive error accumulation. A qualitative analysis in Appendix B.4 empirically confirms this behavior, showing that continuous predictive fusion progressively distorts the memory state and amplifies small predictive biases over time, especially after occlusion. Motivated by this observation, we adopt a reliability-guided activation mechanism that selectively enables predictive prompting. In practice, we first evaluate observation-driven confidence to determine whether the current evidence is sufficiently reliable to rely on observation alone. If the observation-driven confidence

falls below a predefined threshold, we invoke the predictive dynamics to compute a predictive prior and evaluate prediction-observation consistency. Accordingly, we define the reliability signal $\rho_t \in \{0, 1\}$, which determines whether predictive memory is injected into the decoder, as

$$\rho_t = \begin{cases} \mathbb{I}\left[d_t^{\mathrm{feature}} < \tau_{\mathrm{feature}}\right], & \text{if } q_t^{\mathrm{iou}} < \tau_{\mathrm{iou}}, \\ 0, & \text{otherwise}, \end{cases} \tag{9}$$

where the feature-consistency measure $d_t^{\mathrm{feature}}$ is evaluated only when the observation-driven confidence falls below the threshold $q_t^{\mathrm{iou}} < \tau_{\mathrm{iou}}$. Predictive outputs are injected into the decoder as a dense prompt only when both stages of the reliability check are satisfied.

This conditional structure is designed to remain stable even under unreliable observations. Since $d_t^{\mathrm{feature}}$ is invoked only after $q_t^{\mathrm{iou}}$ has already identified the observation as low-confidence, it serves not as a verification of observation correctness, but as a check on whether the prediction has remained consistent with recent observation-driven evidence. When the observation is partially degraded but the prediction still aligns with the target, the discrepancy stays low and predictive prompting is appropriately activated. If the prediction has instead drifted from the target, the discrepancy increases and the gate is suppressed; when the observation is dominated by background or distractors, agreement with the predictive feature is similarly low, preventing spurious activation.

When $\rho_t = 1$, predictive prompting participates in the decoding process and consequently affects the internal state updated by the underlying SAM2 memory mechanism. When $\rho_t = 0$, inference and memory updates rely solely on observation-driven cues, preserving the standard SAM2 update behavior.

**Reliability-Guided Spatial Anchoring.** Beyond the gating decision, when the current observation is already degraded, both prediction and observation may become consistently incorrect, propagating drift through recursive crop updates. To ensure reliable spatial grounding of predictive cues, we adapt the anchoring strategy based on the reliability of the current observation. When the observation is reliable ($q_t^{\mathrm{obj}} \geq \tau_{\mathrm{obj}}$), predictive prompts are aligned using the observation-driven reference from the previous frame, consistent with standard memory propagation. When the observation is unreliable ($q_t^{\mathrm{obj}} < \tau_{\mathrm{obj}}$), we instead anchor predictive prompts to the most recent past frame in which the object was confidently observed. This anchor frame provides a stable geometric reference, preventing noisy or missing observations from distorting the spatial alignment of predictive cues.

As a result, predictive prompting is applied only when the reliability condition is satisfied. In this case, the decoder re-

*Table 1.* Quantitative comparisons with state-of-the-art trackers on **GOT-10k**, **LaSOT**, and **LaSOT**$_{ext}$. **Bold** indicates the best performance and underline denotes the second best.

| Trackers | Source | #Param (M) | GOT-10k | | | LaSOT | | | LaSOT$_{ext}$ | | |
|---|---|---|---|---|---|---|---|---|---|---|---|
| | | | AO | SR$_{0.5}$ | SR$_{0.75}$ | AUC | $P_{\text{norm}}$ | P | AUC | $P_{\text{norm}}$ | P |
| *Supervised VOT methods* | | | | | | | | | | | |
| STARK$_{320}$ (Yan et al., 2021) | ICCV'21 | 47 | 68.8 | 78.1 | 64.1 | 67.1 | 76.9 | 72.2 | 47.8 | 56.2 | 55.2 |
| OSTrack$_{384}$ (Ye et al., 2022) | ECCV'22 | 93 | 73.7 | 83.2 | 70.8 | 71.1 | 81.1 | 77.6 | 50.5 | 61.3 | 57.6 |
| SwinTrack-B$_{384}$ (Lin et al., 2022) | NeurIPS'22 | 91 | 72.4 | 80.5 | 67.8 | 71.3 | - | 76.5 | 49.1 | - | 55.6 |
| ROMTrack$_{384}$ (Cai et al., 2023) | ICCV'23 | 92 | 74.2 | 84.3 | 72.4 | 71.4 | 81.4 | 78.2 | 51.3 | 62.4 | 58.6 |
| SeqTrack-B$_{384}$ (Chen et al., 2023) | CVPR'23 | 89 | 74.5 | 84.3 | 71.4 | 71.5 | 81.1 | 77.8 | 50.5 | 61.6 | 57.5 |
| LoRAT-B$_{378}$ (Lin et al., 2024) | ECCV'24 | 99 | 73.7 | 82.6 | 72.9 | 72.9 | 81.9 | 79.1 | 53.1 | 64.8 | 60.6 |
| EVPTrack$_{384}$ (Shi et al., 2024) | AAAI'24 | 73 | 76.6 | 86.7 | 73.9 | 72.7 | 82.9 | 80.3 | 53.7 | 65.5 | 61.9 |
| HIPTrack$_{384}$ (Cai et al., 2024) | CVPR'24 | 120 | 77.4 | 88.0 | 74.5 | 72.7 | 82.9 | 79.5 | 53.0 | 64.3 | 60.6 |
| AQATrack$_{384}$ (Xie et al., 2024) | CVPR'24 | 72 | 76.0 | 85.2 | 74.9 | 72.7 | 82.9 | 80.2 | 52.7 | 64.2 | 60.8 |
| ARTrackV2$_{384}$ (Bai et al., 2024) | CVPR'24 | 135 | 77.5 | 86.0 | 75.5 | 73.0 | 82.0 | 79.6 | 52.9 | 63.4 | 59.1 |
| ODTrack-B$_{384}$ (Zheng et al., 2024) | AAAI'24 | 93 | 77.0 | 87.9 | 75.1 | 73.2 | 83.2 | 80.6 | 52.4 | 63.9 | 60.1 |
| SUTrack-B$_{384}$ (Chen et al., 2025b) | AAAI'25 | 70 | 79.3 | 88.0 | **80.0** | 74.4 | 83.9 | 81.9 | 52.9 | 63.6 | 60.1 |
| DreamTrack$_{384}$ (Guo et al., 2025) | CVPR'25 | 87 | 78.3 | 87.9 | 76.6 | 75.0 | 84.2 | 81.7 | 54.5 | 65.3 | 61.1 |
| MCITrack$_{384}$ (Kang et al., 2025) | AAAI'25 | 88 | 77.9 | 88.2 | 76.8 | **75.3** | **85.6** | **83.3** | 54.6 | 65.7 | 62.1 |
| *Zero-shot SAM2-based methods* | | | | | | | | | | | |
| SAMURAI-B (Yang et al., 2024) | arXiv'24 | 81 | 79.6 | **90.8** | 72.9 | 70.7 | 78.7 | 76.2 | 57.5 | 69.3 | 67.1 |
| SAM2.1-B (Ravi et al., 2025) | ICLR'25 | 81 | 77.9 | 88.6 | 71.5 | 66.0 | 73.5 | 71.0 | 55.5 | 67.2 | 64.6 |
| DAM4SAM-B (Videnovic et al., 2025) | CVPR'25 | 81 | 78.1 | 88.5 | 70.9 | 73.3 | 81.3 | 78.8 | 58.6 | 69.4 | 68.2 |
| HiM2SAM-B (Chen et al., 2025a) | PRCV'25 | 81 | 78.1 | 88.9 | 71.7 | 73.4 | 81.7 | 79.5 | 59.3 | 71.2 | 69.4 |
| Ours-B | ICML'26 | 93 | **79.7** | 90.6 | 72.9 | 74.5 | 82.9 | 80.7 | **60.9** | **73.1** | **71.4** |

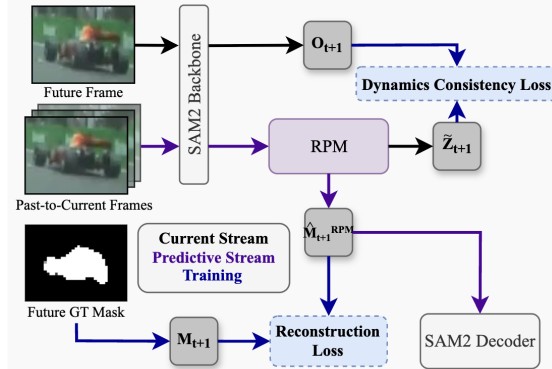

*Figure 2.* Predictive memory training with SAM2 for feature extraction and RPM for latent prediction.

ceives an additional predictive prompt $\mathbf{p}_t^{\text{RPM}}$, which is constructed from the decoder-aligned dense prediction $\hat{\mathbf{M}}_t^{\text{RPM}}$ by spatial alignment and subsequent encoding through the SAM2 prompt encoder, following the strategy described above. The resulting decoder input is given by

$$\tilde{\mathbf{o}}_t = \mathbf{o}_t + \rho_t \cdot \mathbf{p}_t^{\text{RPM}}, \tag{10}$$

ensuring that predictive memory influences perception only under reliable conditions and is consistently grounded on a stable spatial reference.

### 3.6. Loss Function

Our training objective supervises the future prediction module using three complementary loss components: a latent prediction loss and a mask prediction loss composed of binary cross-entropy and Dice losses, as illustrated in Fig. 2.

**Dynamics Consistency Loss.** To learn predictive dynamics that remain consistent with observation-driven memory, we supervise the predictive latent representation ($\tilde{z}_{t+1}$) using the observation-driven feature ($o_{t+1}$):

$$\mathcal{L}_{\text{dyn}} = \|\tilde{\mathbf{z}}_{t+1} - \mathbf{o}_{t+1}\|_2^2. \tag{11}$$

This loss encourages the state-space dynamics to evolve latent states that are aligned with the observed memory representation.

**Reconstruction Loss.** In addition to latent supervision, we directly supervise the mask logits $\hat{M}_{t+1}^{\text{RPM}}$ predicted by the future prediction module using the ground-truth binary mask $M_{t+1}$:

$$\mathcal{L}_{\text{mask}} = \mathcal{L}_{\text{BCE}}(\hat{\mathbf{M}}_{t+1}^{\text{RPM}}, \mathbf{M}_{t+1}) + \mathcal{L}_{\text{Dice}}(\hat{\mathbf{M}}_{t+1}^{\text{RPM}}, \mathbf{M}_{t+1}), \tag{12}$$

where $\hat{M}_{t+1}^{\text{RPM}}$ denotes the predictive mask logit produced by the RPM module and $M_{t+1}$ denotes the ground-truth binary mask.

The final training objective is defined as

$$\mathcal{L} = \lambda_{\text{dyn}}\mathcal{L}_{\text{dyn}} + \lambda_{\text{mask}}\mathcal{L}_{\text{mask}}. \tag{13}$$

No loss is applied to the reliability signal $\rho_t$, which is computed deterministically and used solely to regulate perceptual fusion by controlling predictive prompt activation.

## 4. Experiments

### 4.1. Experimental Settings

We adopt a zero-shot evaluation protocol, where the future prediction module is trained on video segmentation

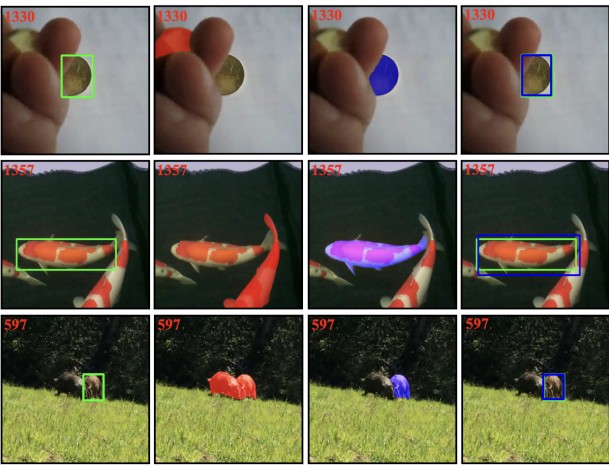

Input(Ground Truth)  SAM2.1 without RPM  SAM2.1 with RPM  Output

*Figure 3.* Qualitative comparison of segmentation results. From left to right: ground-truth input, SAM2.1 without reliability guidance, SAM2.1 with reliability guidance, and the proposed method.

data while evaluation is performed on video object tracking benchmarks without any task-specific fine-tuning. Our framework is built upon a pretrained SAM2.1, and to improve the robustness of the memory bank, we employ a memory update strategy following HiM2SAM (Chen et al., 2025a). On top of this foundation, we introduce the proposed RPM module. During training, all pretrained parameters of SAM2.1 are kept frozen, and only the RPM module is optimized. This design enables the model to learn future-aware and reliability-sensitive representations while preserving the original segmentation capability of the SAM2.1 backbone. Unless stated otherwise, all experiments follow this configuration.

We train our model using SA-V (Ravi et al., 2025), MOSE (Ding et al., 2023), and YouTube-VOS 2019 (Xu et al., 2018). SA-V provides long video sequences with complex temporal dynamics, MOSE emphasizes heavy occlusion and distractor interference, and YouTube-VOS 2019 improves generalization across diverse object categories and motion patterns.

We evaluate our method on LaSOT (Fan et al., 2019), $\text{LaSOT}_{ext}$ (Fan et al., 2021), and GOT-10k (Huang et al., 2021), where LaSOT and $\text{LaSOT}_{ext}$ are long-term single-object tracking benchmarks with extended sequences and significant appearance variations. GOT-10k evaluates generalization under a strict train-test class separation.

## 4.2. Quantitative and Qualitative Analysis

We evaluate the overall tracking performance of the proposed method on multiple standard benchmarks using the same pretrained SAM2.1 backbone for all SAM2-based experiments while retaining each method's original mem-

*Table 2.* Ablation study on the future prompt usage strategies. Latency denotes the average per-frame inference time, and Overhead indicates the relative latency increase compared to SAM2.1.

| Method | AUC | Latency(ms) | Overhead(%) |
|---|---|---|---|
| SAM2.1 | 66.0 | 278.25 | - |
| + RPM (w/o Reliability-Guided Fusion) | 70.5 | 340.05 (+61.80) | +22.2 |
| + RPM (w/ Reliability-Guided Fusion) | **74.5** | **280.54 (+2.29)** | **+0.82** |

ory update strategy. As shown in Table 1, our approach consistently achieves competitive performance with strong supervised trackers and recent zero-shot SAM2-based methods across public datasets. Notably, the performance gains are more pronounced on long-term tracking benchmarks, where maintaining object identity over extended temporal horizons is particularly challenging. Qualitative results further illustrate the effectiveness of reliability-guided predictive memory. As shown in Fig. 3, observation-driven trackers are prone to drift after occlusion or abrupt motion, frequently failing to re-localize the target upon reappearance. In contrast, our method achieves more stable object localization and segmentation by selectively leveraging predictive memory when observation quality degrades. This behavior indicates that RPM effectively balances observation-driven perception and predictive reasoning, enabling robust recovery under challenging conditions. Overall, both quantitative results and qualitative evidence demonstrate that explicitly regulating predictive memory is critical for robust online video perception. Rather than relying on unconditional future estimation, the proposed reliability-guided strategy allows predictive cues to contribute only when they are reliable and most beneficial.

### 4.3. Ablation Study

**Future Prompt Integration.** We analyze different strategies for using future prompts in terms of both tracking performance and computational efficiency on the LaSOT benchmark. As shown in Table 2, using future prediction prompts without reliability regulation (*RPM w/o Reliability-Guided Fusion*) results in inferior performance and substantial latency overhead, since predictive prompts are generated at every frame regardless of observation quality. Under uncertain conditions, such unregulated prompt usage can overwhelm observation-driven cues, leading to inefficient and potentially unstable inference. In contrast, when future prompts are controlled by the proposed reliability-guided fusion mechanism (*RPM w/ Reliability-Guided Fusion*), tracking performance improves consistently while incurring only negligible additional latency. Importantly, the observed latency differences are primarily determined by whether predictive prompts are generated at a given frame, rather than

*Table 3.* Ablation of reliability indicator components.

| $q_{iou}$ | $q_{obj}$ | $d_{feature}$ | AUC | $P_{norm}$ | P |
|:---:|:---:|:---:|:---:|:---:|:---:|
| × | × | × | 70.5 | 78.7 | 76.6 |
| ✓ | × | × | 73.8 | 82.0 | 79.8 |
| ✓ | ✓ | × | 73.9 | 82.1 | 79.9 |
| ✓ | × | ✓ | 74.0 | 82.2 | 80.1 |
| ✓ | ✓ | ✓ | **74.5** | **82.9** | **80.7** |

*Table 4.* Ablation study on spatial anchoring strategies.

| Anchoring Strategy | AUC | $P_{norm}$ | P |
|:---|:---:|:---:|:---:|
| Previous-frame | 73.8 | 82.1 | 79.8 |
| Last reliable-frame | 74.1 | 82.4 | 80.2 |
| Reliability-guided | **74.5** | **82.9** | **80.7** |

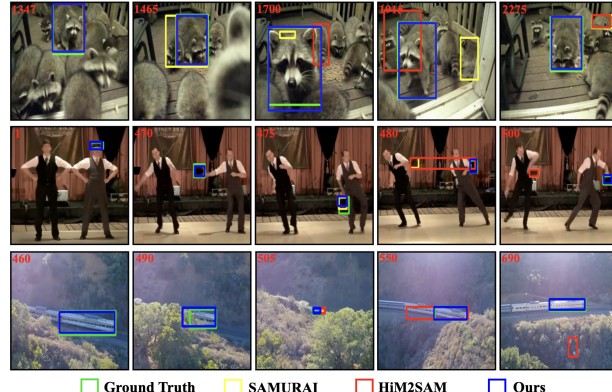

*Figure 4.* Qualitative comparison of tracking results between current observation-based and predictive memory-based trackers.

*Table 5.* Occlusion-aware analysis stratified by occlusion duration. Performance is measured by mean IoU over the first 5 frames after occlusion ends.

| Trackers | Short(<5) | Medium(5–20) | Long(>20) |
|:---|:---:|:---:|:---:|
| SAMURAI | 0.498 | 0.421 | 0.311 |
| HiM2SAM | 0.552 | 0.482 | 0.384 |
| Ours | **0.577** | **0.517** | **0.421** |

by the prompt fusion operation itself, as discussed in Appendix B.3. These results highlight that reliability-aware prompt activation is critical for both robust tracking and the efficient integration of predictive memory into SAM2.

**Reliability Indicators.** We analyze the reliability signal from the perspective of the contribution of individual indicators to prompt control on the LaSOT benchmark. Table 3 evaluates different indicator combinations for future prompt activation with additional implementation details and extended analyses provided in Appendix B.2. Without a reliability signal, the model degenerates to unconditional prompt usage, resulting in lower performance. While a single indicator yields only marginal gains, combining multiple indicators leads to consistent improvements. The best performance is obtained when object confidence, IoU-based prediction quality, and feature-level discrepancy are jointly considered, indicating that effective and safe prompt activation requires complementary cues capturing object presence, prediction accuracy, and appearance consistency. This analysis highlights that reliability estimation is a central design component rather than an auxiliary heuristic, and that its effectiveness fundamentally relies on the interaction of multiple, semantically distinct cues.

**Spatial Anchoring.** We analyze the spatial anchoring strategy introduced in Section 3.5 on the LaSOT benchmark. Since predictive prompts must be spatially aligned with the target before decoding, the choice of spatial reference determines whether unreliable cues propagate into subsequent memory updates. Table 4 compares three anchoring variants under the same reliability-guided activation: the immediately preceding frame, the last past frame with reliable object confidence, and the proposed reliability-guided anchoring that adaptively selects between the two based on $q_t^{obj}$. Anchoring to the preceding frame consistently underperforms, particularly on precision-oriented metrics, as un-

reliable references propagate through recursive crop updates and distort predictive guidance. Anchoring to the last reliable frame breaks this drift-crop feedback loop and yields moderate gains, but applies the same reference regardless of the current observation, ignoring recent evidence when it is reliable. The proposed reliability-guided anchoring resolves this trade-off by relying on the preceding frame when observations remain reliable and reverting to the last reliable frame only under degraded observations, achieving the best performance across all metrics. These results show that the spatial reference itself contributes to predictive prompting, and that its adaptation to observation reliability is essential for stable spatial grounding.

### 4.4. Detailed Analysis

**Occlusion Duration.** We evaluate robustness under short-term, medium-term, and long-term occlusions on the LaSOT benchmark to examine how our method behaves under varying occlusion durations. Occlusion length is measured using LaSOT frame-level annotations and categorized as short-term (<5 frames), medium-term (5-20 frames), and long-term (>20 frames) based on the number of consecutive fully occluded frames. As shown in Fig. 4 and Table 5, our method consistently achieves the strongest performance across all occlusion regimes, indicating robust behavior under object disappearance. Notably, the performance gap becomes substantially larger in the long-term setting, where

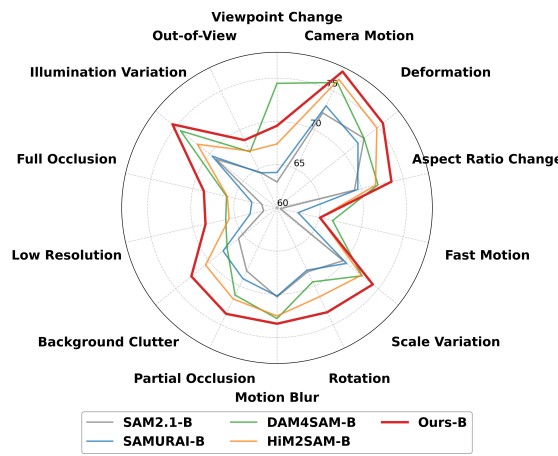

SAM2.1-B    DAM4SAM-B    Ours-B
SAMURAI-B    HiM2SAM-B

*Figure 5.* Quantitative comparison of attribute-wise AUC.

*Table 6.* $\mathcal{J}\&\mathcal{F}$ comparison of SAM2.1 and RPM on seven VOS benchmarks under the same configuration.

| Model | SA-V val | SA-V test | LVOS v1 val | LVOS v2 val |
|---|---|---|---|---|
| SAM2.1 | 78.0 | 77.7 | 79.7 | 83.1 |
| **Ours** | **79.5** | **79.8** | **81.6** | **83.9** |

| Model | MOSE val | DAVIS2017 val | YTVOS2019 val |
|---|---|---|---|
| SAM2.1 | 73.8 | 90.0 | 88.3 |
| **Ours** | **74.6** | **90.7** | **88.5** |

the target remains unobservable for extended periods. In such cases, relying on observation-driven memory alone is often insufficient to preserve object identity across prolonged gaps. By selectively introducing predictive cues only when observation reliability degrades, while preventing speculative information from being permanently written into memory, the proposed reliability-guided predictive prompting enables more stable identity preservation and more accurate recovery after extended occlusion. This long-term robustness emerges from repeatedly applying a local decision rule that activates or suppresses predictive prompts at each step, rather than from explicit long-horizon prediction.

**Attribute-wise Analysis.** We examine the behavior of the proposed method across different tracking attributes on the LaSOT benchmark. We report attribute-wise AUC over 14 standard attributes to assess how reliability-guided predictive memory operates under diverse motion and visibility conditions. Fig. 5 summarizes these results in a quantitative visualization. Overall, the proposed method yields consistent performance gains across the majority of attributes, with particularly pronounced improvements under degraded visibility conditions. In partial occlusion scenarios, our method improves upon HiM2SAM by approximately 1.9% AUC, indicating more reliable target recovery when the object is intermittently observable. Under full occlusion, our approach achieves a 2.7% AUC improvement over DAM4SAM, demonstrating the effectiveness of reliability-guided predictive activation in the absence of visual evidence. Similarly, in out-of-view cases, our method outperforms HiM2SAM by 1.4% AUC, suggesting improved stability during long-term target disappearance and subsequent re-entry. A detailed quantitative analysis with full results for all attributes is provided in Appendix B.1.

**Generalization Beyond Tracking.** We further examine whether RPM generalizes beyond visual object tracking by

evaluating the same trained model on seven standard video object segmentation (VOS) benchmarks, covering SA-V val/test (Ravi et al., 2025), LVOS v1/v2 val (Hong et al., 2023; 2025), MOSE val (Ding et al., 2023), DAVIS2017 val (Pont-Tuset et al., 2017), and YTVOS2019 val (Xu et al., 2018). As shown in Table 6, RPM consistently improves $\mathcal{J}\&\mathcal{F}$ over the SAM2.1 baseline across all benchmarks, indicating that the reliability-guided mechanism transfers to segmentation tasks without architectural redesign. The improvements are most pronounced on long-form benchmarks such as SA-V and LVOS, which is consistent with the design rationale of RPM: predictive memory becomes most beneficial when observation reliability fluctuates over extended temporal horizons. This generalization follows naturally from the design of RPM: the reliability signal is derived from intrinsic properties of the perception backbone, namely observation confidence and prediction–observation consistency, rather than from task-specific supervision. As a result, the same gating mechanism operates consistently across both tracking and segmentation regimes. Additional analyses on cross-task threshold stability and computational efficiency in the segmentation regime are provided in Appendices B.2 and B.3.

## 5. Conclusion

We propose a framework for incorporating predictive dynamics into online video perception under uncertainty, called RPM. We identify a fundamental reliability mismatch in existing predictive memory designs, where unregulated predictive updates contaminate internal states and result in irreversible identity drift, particularly after occlusion. RPM formulates video memory as a dynamic latent process whose evolution is explicitly controlled by reliability. By combining state-space predictive dynamics with a reliability-aware control signal, the proposed approach regulates the perceptual influence of predictive priors and prevents unreliable updates from corrupting long-term memory. We instantiate RPM on a SAM2-based foundational video model and demonstrate consistent improvements on challenging long-term tracking benchmarks. These results show that predictive memory is effective only when its influence is explicitly regulated by reliability, and point to a general principle for robust online video perception.

## Impact Statement

This paper presents work aimed at improving the robustness of online video perception systems through reliability-guided predictive memory. Such advances can benefit applications including autonomous driving, robotics, and assistive video understanding. We do not foresee any consequences that require specific highlighting beyond those typical of advances in machine learning.

## Acknowledgments

S.M. Yoon is supported by the National Research Foundation of Korea (No. RS-2025-00555827), and the Institute of Information & Communications Technology Planning & Evaluation (IITP) grant (No. RS-2025-02219317, AI Star Fellowship (Kookmin University)) funded by the Korean government (MSIT). M. Kim is supported by the MSIT, Korea, under the National Program for Excellence in SW (No. 2022-0-00964), supervised by the IITP in 2026.

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

# A. Training and Implementation Details

### A.1. Training Configuration

The Reliability-Guided Predictive Memory (RPM) module is optimized using the AdamW optimizer (Loshchilov & Hutter, 2019) with a base learning rate of $1 \times 10^{-4}$. Training is conducted with a batch size of 1 and a tracklet length of 8 frames. We follow the standard SAM2.1 training protocol and train the RPM model for 50 epochs. Automatic mixed precision with bfloat16 is used throughout training. The RPM module is trained using supervision applied exclusively to future predictions. The training objective consists of a latent dynamics consistency loss (MSE) and a future mask prediction loss composed of binary cross-entropy (BCE) and Dice loss. All three losses are equally weighted with a coefficient of 1.0, while conventional segmentation, IoU, and classification losses are disabled during training. Training samples are drawn from a mixture of video segmentation datasets, including SA-V, MOSE, and YouTube-VOS 2019, with sampling multipliers of 1.0, 7.0, and 3.0, respectively, to balance object diversity and motion complexity.

### A.2. Frozen Backbone Training Protocol

During training, all parameters of the pretrained SAM2.1 backbone, including the image encoder, memory attention, and memory encoder, are frozen. Only the parameters introduced by the RPM framework are updated. Specifically, gradient updates are enabled for the predictive dynamics backbone (state-space dynamics based on Mamba), normalization layers, projection MLPs, decoder-side adaptation layers, and logit heads associated with RPM, while all remaining SAM2.1 parameters are kept fixed. This strategy allows RPM to learn predictive dynamics and reliability-aware control without altering the original perception and memory representations of the foundation model.

### A.3. Reliability-guided Predictive Memory Configuration

Table 7 summarizes the architectural components and hyperparameter settings of the Reliability-Guided Predictive Memory (RPM) used across all experiments.

*Table 7.* Architectural and hyperparameter settings of Reliability-Guided Predictive Memory (RPM).

| Component | Setting |
|---|---|
| Latent dimension | 256 |
| Prediction horizon | 1-step (next frame) |
| Input image resolution | $224 \times 224$ |
| Input feature resolution | $14 \times 14$ |
| Tracklet length | 8 frames |
| Temporal embedding | Sinusoidal (fixed) |
| Temporal embedding scale | 0.1 |
| Predictive dynamics backbone | Mamba2 |
| Dynamics depth | 24 layers |
| State dimension ($d_{state}$) | 128 |
| Convolution width ($d_{conv}$) | 4 |
| Expansion ratio | 2 |
| MLP expansion ratio | 4.0 |
| Decoder-aligned output resolution | $64 \times 64$ |
| Decoder adaptation | Upsample + Conv + LN + GELU |
| Prediction head | $1 \times 1$ Conv |
| Feature discrepancy threshold ($\tau_{\text{feature}}$) | 0.3 |
| Object confidence threshold ($\tau_{\text{obj}}$) | 0.3 |
| IoU reliability threshold ($\tau_{\text{iou}}$) | 0.4 |
| Target crop scale ($\alpha_{\text{crop}}$) | 2.0 |
| Objectness scaling factor ($\lambda_o$) | 1.0 |
| IoU scaling factor ($\lambda_i$) | 1.0 |
| Predictive activation | Reliability-gated |

The RPM module is designed as a compact and conditionally executed predictive component that operates on short-term tracklets and generates one-step-ahead latent predictions for online video perception. We adopt a one-step prediction horizon to align predictive dynamics with the frame-by-frame memory update process used by memory-centric video models. Since

the internal memory state is updated at every time step and recursively propagated over long sequences, even single-step predictive guidance can exert long-horizon influence through repeated state evolution. Under partial observability, this design provides a temporally local yet structurally consistent mechanism for shaping memory dynamics without introducing uncontrolled long-range speculation.

*Table 8.* Correspondence between the abstract state-space formulation (Eqs. (4)–(6)) and its Mamba2 instantiation.

| Symbol | Role | Mamba2 instantiation |
|---|---|---|
| $\bar{\mathbf{A}}$ | State transition | Input-dependent, modulated by $\Delta$ |
| $\bar{\mathbf{B}}$ | Input projection | Selective, input-dependent |
| $\mathbf{C}$ | Output projection | Selective output projection to latent space |
| $\Delta$ | Step size | Learned, input-conditioned |

Predictive dynamics are implemented with a 24-layer Mamba2 backbone, which provides an efficient and expressive realization of the state-space formulation introduced in Section 3.2. The abstract operators $\bar{\mathbf{A}}$, $\bar{\mathbf{B}}$, and $\mathbf{C}$ in Eqs. (4)–(6) describe how latent states evolve, integrate new evidence, and project onto the predictive feature space, respectively. As summarized in Table 8, Mamba2 instantiates these operators through input-conditioned, discretized, and head-wise state-space parameters, where the learned step size $\Delta$ governs the selectivity of $\bar{\mathbf{A}}$ and $\bar{\mathbf{B}}$ at each time step. This selective parameterization enables rich temporal modeling in high-dimensional feature spaces while preserving the linear-time efficiency of state-space recurrence. Architectural depth and width primarily increase the expressiveness of the predictive dynamics rather than the prediction horizon itself, allowing predictive capacity to grow without extending the temporal rollout.

All thresholds and architectural choices are selected to balance predictive capacity and stability under occlusion. We empirically observe that performance is robust to moderate variations of these values. Unless otherwise specified, these settings are fixed across all datasets and evaluation protocols. The reliability thresholds $\tau_{\mathrm{iou}}$, $\tau_{\mathrm{obj}}$, and $\tau_{\mathrm{feature}}$ are fixed to a single global configuration, selected once on validation data and shared across all datasets and SAM2-based backbones.

## B. Additional Experimental Details

### B.1. Attribute-wise AUC Analysis on LaSOT

We evaluate attribute-wise AUC over 14 standard LaSOT attributes, covering dynamic motion factors such as viewpoint change (VC), camera motion (CM), deformation (DEF), aspect ratio change (ARC), fast motion (FM), scale variation (SV), rotation (ROT), and motion blur (MB) as well as degraded visibility conditions, including partial occlusion (POC), background clutter (BC), low resolution (LR), full occlusion (FOC), illumination variation (IV), and out-of-view (OV). Table 9 compares SAM2.1, SAMURAI, DAM4SAM, HiM2SAM, and the proposed method under a unified SAM2-based backbone.

*Table 9.* Attribute-wise AUC (%) comparison on LaSOT using SAM2-based models. Attributes are grouped into *Dynamic Motion* and *Degraded Visibility*. **Bold** indicates the best performance, and underline denotes the second-best result.

| | LaSOT | | | | | | | | | | | | | |
|---|---|---|---|---|---|---|---|---|---|---|---|---|---|---|
| | Dynamic Motion | | | | | | | | Degraded Visibility | | | | | |
| **Trackers** | VC | CM | DEF | ARC | FM | SV | ROT | MB | POC | BC | LR | FOC | IV | OV |
| SAM2.1-B | 63.0 | 72.2 | 72.9 | 69.2 | 60.4 | 69.8 | 68.2 | 70.3 | 68.1 | 65.7 | 61.6 | 61.8 | 69.0 | 64.6 |
| SAMURAI-B | 64.1 | 73.1 | 72.0 | 69.6 | 62.5 | 70.3 | 68.0 | 70.2 | 69.1 | 68.0 | 63.2 | 63.0 | 69.6 | 64.5 |
| DAM4SAM-B | **74.4** | 76.1 | 72.9 | 72.0 | **66.6** | 72.6 | 69.5 | 72.8 | 71.2 | 67.4 | 66.1 | 66.0 | 74.3 | 67.2 |
| HiM2SAM-B | 67.4 | 76.5 | 74.8 | 71.7 | 65.1 | 72.5 | 71.4 | 72.5 | 71.7 | 70.6 | 65.7 | 65.9 | 71.8 | 67.3 |
| Ours-B | 69.5 | **77.5** | **75.7** | **73.6** | 65.1 | **74.2** | **73.4** | **73.4** | **73.6** | **72.7** | **68.5** | **68.7** | **75.5** | **68.7** |
| **Difference** | -4.9 | **1.0** | **0.9** | **1.6** | -1.5 | **1.6** | **2.0** | **0.6** | **1.9** | **2.1** | **2.4** | **2.7** | **1.2** | **1.4** |

Overall, the proposed method achieves consistent improvements across most attributes. While gains under *Dynamic Motion* are competitive, larger improvements are observed under *Degraded Visibility*. In particular, substantial gains are achieved

under partial occlusion (POC), full occlusion (FOC), and out-of-view (OV), with relative improvements of +1.9%, +2.7%, and +1.4% AUC, respectively. These scenarios correspond to conditions where observation-driven evidence is incomplete or entirely missing. The pronounced improvements under these attributes reflect the design of Reliability-Guided Predictive Memory (RPM), which selectively activates predictive cues only when observation reliability degrades, thereby preventing unreliable predictions from contaminating memory and enabling more stable tracking through periods of visual absence.

### B.2. Parameter Analysis of Reliability-Guided Predictive Memory

We examine the robustness of the resulting gating behavior to threshold choices on the LaSOT benchmark. Table 10 evaluates the stability of the proposed reliability-gated mechanism under variations of key thresholds and hyperparameters. Across a broad range of IoU confidence, object confidence, feature discrepancy thresholds, and search region expansion ratios, performance remains largely stable. This behavior indicates that the proposed reliability signal does not rely on finely tuned thresholds, but instead induces a robust decision regime in which prompt activation is governed by relative consistency rather than absolute parameter values. Taken together, these results demonstrate that the proposed reliability-guided design not only improves performance, but also provides a stable and deployable mechanism for controlling predictive memory under uncertainty.

*Table 10.* Threshold sensitivity analysis of reliability-gated predictive memory on LaSOT benchmark.

| $q_{iou}$ | AUC | $P_{norm}$ | P |
|---|---|---|---|
| 0.3 | 74.4 | 82.8 | 80.6 |
| 0.4 | **74.5** | **82.9** | **80.7** |
| 0.5 | 74.3 | 82.5 | 80.3 |
| 0.7 | 74.2 | 82.4 | 80.2 |

**(a)** $q_{iou}$

| $q_{obj}$ | AUC | $P_{norm}$ | P |
|---|---|---|---|
| 0.0 | 74.0 | 82.3 | 80.0 |
| 0.1 | 74.3 | 82.6 | 80.4 |
| 0.2 | 74.0 | 82.2 | 80.0 |
| 0.3 | **74.5** | **82.9** | **80.7** |

**(b)** $q_{obj}$

| $d_{feature}$ | AUC | $P_{norm}$ | P |
|---|---|---|---|
| 0.2 | 74.0 | 82.3 | 80.1 |
| 0.3 | **74.5** | **82.9** | **80.7** |
| 0.4 | 74.1 | 82.5 | 80.2 |
| 0.5 | 74.1 | 82.3 | 80.1 |

**(c)** $d_{feature}$

| $\alpha_{crop}$ | AUC | $P_{norm}$ | P |
|---|---|---|---|
| 1.5 | 74.4 | 82.7 | 80.5 |
| 2.0 | **74.5** | **82.9** | **80.7** |
| 2.5 | 74.5 | 82.8 | 80.6 |
| 3.0 | 74.3 | 82.6 | 80.4 |

**(d)** $\alpha_{crop}$

This stability further extends to the segmentation regime. Table 11 reports $\mathcal{J}\&\mathcal{F}$ on SA-V val and LVOS v2 val under the same set of threshold variations. The gating behavior exhibits a comparable robustness profile, with performance varying only within a narrow band across the tested ranges. This invariance across tracking and segmentation suggests that the reliability signal captures a broadly task-agnostic notion of perceptual uncertainty, though tasks with substantially different observation statistics may benefit from light recalibration.

*Table 11.* Threshold sensitivity analysis of reliability-gated predictive memory on VOS benchmarks.

| $q_{iou}$ | SA-V val | LVOS v2 val |
|---|---|---|
| 0.3 | 79.3 | 83.8 |
| 0.4 | **79.5** | **83.9** |
| 0.5 | **79.5** | 83.7 |
| 0.7 | 79.1 | 83.2 |

**(a)** $q_{iou}$

| $q_{obj}$ | SA-V val | LVOS v2 val |
|---|---|---|
| 0.0 | 79.2 | 83.7 |
| 0.1 | 79.3 | 83.5 |
| 0.2 | 79.2 | 83.6 |
| 0.3 | **79.5** | **83.9** |

**(b)** $q_{obj}$

| $d_{feature}$ | SA-V val | LVOS v2 val |
|---|---|---|
| 0.2 | 79.4 | 83.7 |
| 0.3 | **79.5** | **83.9** |
| 0.4 | 79.3 | 83.5 |
| 0.5 | 78.9 | 83.4 |

**(c)** $d_{feature}$

| $\alpha_{crop}$ | SA-V val | LVOS v2 val |
|---|---|---|
| 1.5 | **79.5** | **84.0** |
| 2.0 | **79.5** | 83.9 |
| 2.5 | 79.4 | 83.9 |
| 3.0 | 79.1 | 83.5 |

**(d)** $\alpha_{crop}$

### B.3. Computational Overhead

We report the computational overhead introduced by RPM, evaluated on the LaSOT benchmark using the SAM2-Base configuration, with latency measured as the average inference time per frame on a single NVIDIA A100 GPU, as reported in Table 2. Importantly, the measured latency difference between the reliability-guided and unguided variants does not stem from decoder-level integration costs. Instead, it is dominated by whether predictive prompts are generated at a given frame.

Without reliability guidance, predictive prompts are generated at every frame regardless of observation quality, resulting in a consistent and substantial computational overhead. In contrast, RPM triggers predictive prompt generation only when the reliability criterion is satisfied, leading to sparse activation over the benchmark. As quantified in Table 12, the predictive gate is activated in only 3.71% of frames on LaSOT and 4.89% on LaSOT$_{ext}$. These activations occur under two types of conditions. First, predictive prompting is invoked when the target becomes fully unobservable, including full-occlusion and out-of-view cases. Second, it may also be activated under low-confidence observations, where the target remains partially visible but observation reliability falls below the confidence threshold. Consequently, the activation rate exceeds the proportion of fully unobservable frames, reflecting that RPM is guided by reliability signals rather than by binary visibility annotations alone.

As a result, the additional computation associated with predictive memory is incurred only when observation reliability degrades, rather than whenever a target is fully occluded. The reported latency therefore reflects an average cost amortized over all frames, yielding a substantially smaller measured overhead compared to unconditional predictive prompting. As shown in Table 2, the additional cost introduced by RPM remains negligible relative to the backbone and decoder, demonstrating that reliability-guided predictive memory can be integrated into SAM2 without sacrificing efficiency. This overhead corresponds to the empirical activation behavior observed on standard benchmarks, and the per-frame cost naturally scales with the activation rate under sequences with substantially more frequent occlusion.

*Table 12.* Statistics of reliability-guided predictive gate activation under different observation conditions. RPM activates predictive memory sparsely based on observation reliability rather than binary visibility cues. Target unobservability denotes frames in which the target is fully unobservable, including full-occlusion and out-of-view cases. Low-confidence observation denotes frames where the target remains partially visible but observation reliability falls below the confidence threshold. Activation rate denotes the percentage of frames where the predictive gate $\rho_t$ is activated over the entire sequence.

| Dataset | Target Unobservability (%) | | Low-Confidence Observation (%) | Activation Rate (%) |
|---|---|---|---|---|
| | Full-Occlusion | Out-of-View | | |
| LaSOT | 1.14 | 1.54 | 1.03 | 3.71 |
| LaSOT$_{ext}$ | 3.23 | 0.28 | 1.38 | 4.89 |

This efficiency profile also generalizes to the segmentation setting. Table 13 reports the predictive gate activation rate and per-frame latency on SA-V val and LVOS v2 val under the SAM2.1-B configuration. Predictive prompts are activated on only a small fraction of frames (1.68% and 1.92%, respectively), incurring negligible latency overheads of +0.40% and +0.45% relative to the SAM2.1 baseline. The reduced activation rates compared to LaSOT and LaSOT$_{ext}$ are consistent with the lower incidence of full-occlusion and out-of-view events in standard VOS benchmarks, and the per-frame overhead scales proportionally; absolute baseline latency varies across datasets due to differences in average frame size and sequence length. These results demonstrate that RPM achieves consistent efficiency across both tracking and segmentation regimes without task-specific adaptation.

*Table 13.* Activation rate denotes the percentage of frames where the predictive gate $\rho_t$ is activated, and Overhead indicates the relative latency increase over the SAM2.1 baseline.

| Dataset | Activation Rate (%) | Latency (ms) | Overhead (%) |
|---|---|---|---|
| SA-V val | 1.68 | 259.7 (+1.04) | +0.40 |
| LVOS v2 val | 1.92 | 263.1 (+1.18) | +0.45 |

## B.4. Quantitative and Qualitative Failure Analysis of Continuous Predictive Fusion

We compare the proposed binary reliability-guided gating against a learned soft-gating variant on LaSOT. The soft gate takes the three reliability signals as input and is implemented as a lightweight MLP with sigmoid output, isolating the effect of the gating rule itself. As shown in Table 14, soft-gating improves over unconditional injection but remains substantially below the binary counterpart. We attribute this gap to an intrinsic limitation of continuous fusion: any non-zero blending weight admits predictive error into recursive memory updates, whereas binary suppression eliminates predictive influence outright under low reliability.

*Table 14.* Comparison of fusion strategies on LaSOT, including unconditional injection, learned soft-gating, and the proposed binary reliability-guided gating.

| Method | AUC |
|---|---|
| SAM2.1-B | 66.0 |
| Always-on Prediction Injection | 70.5 |
| Soft-Gating (Learned Scalar) | 72.3 |
| Reliability-Guided Binary Gating | **74.5** |

To verify that this propagation indeed occurs in the soft-gating variant, we examine the distribution of the learned scalar gate $\alpha$ on LaSOT. As reported in Table 15, only 5.6% of frames fall into the strong-suppression range ($\alpha < 0.2$), while

the majority (77.7%) lie in the intermediate range ($0.2 \leq \alpha < 0.8$). The learned gate therefore attenuates rather than fully suppresses unreliable predictions, leaving residual influence that compounds through recursive memory updates. Binary suppression, in contrast, deterministically removes the predictive contribution under low reliability, preventing such residual error from entering the memory state in the first place.

*Table 15.* Distribution of the learned scalar gate $\alpha$ across LaSOT frames under the soft-gating variant. Most activations concentrate in the intermediate range, indicating that soft-gating attenuates rather than fully suppresses unreliable predictions.

| Method | $\alpha < 0.2$ | $0.2 \leq \alpha < 0.5$ | $0.5 \leq \alpha < 0.8$ | $\alpha \geq 0.8$ |
|---|---|---|---|---|
| Continuous Fusion | 5.6% | 32.6% | 45.1% | 16.7% |

The downstream consequence of this residual propagation is illustrated qualitatively in Fig. 6. Even weakly weighted predictive prompts introduce subtle but persistent biases into the internal memory state; although these biases do not trigger immediate failure, they are gradually committed to memory and compound over time, eventually leading to severe tracking drift after occlusion. Lowering the blending weight does not eliminate this behavior but merely postpones the onset of failure. Explicit suppression under low reliability, on the other hand, prevents unreliable predictions from being written into memory, thereby blocking error accumulation at its source.

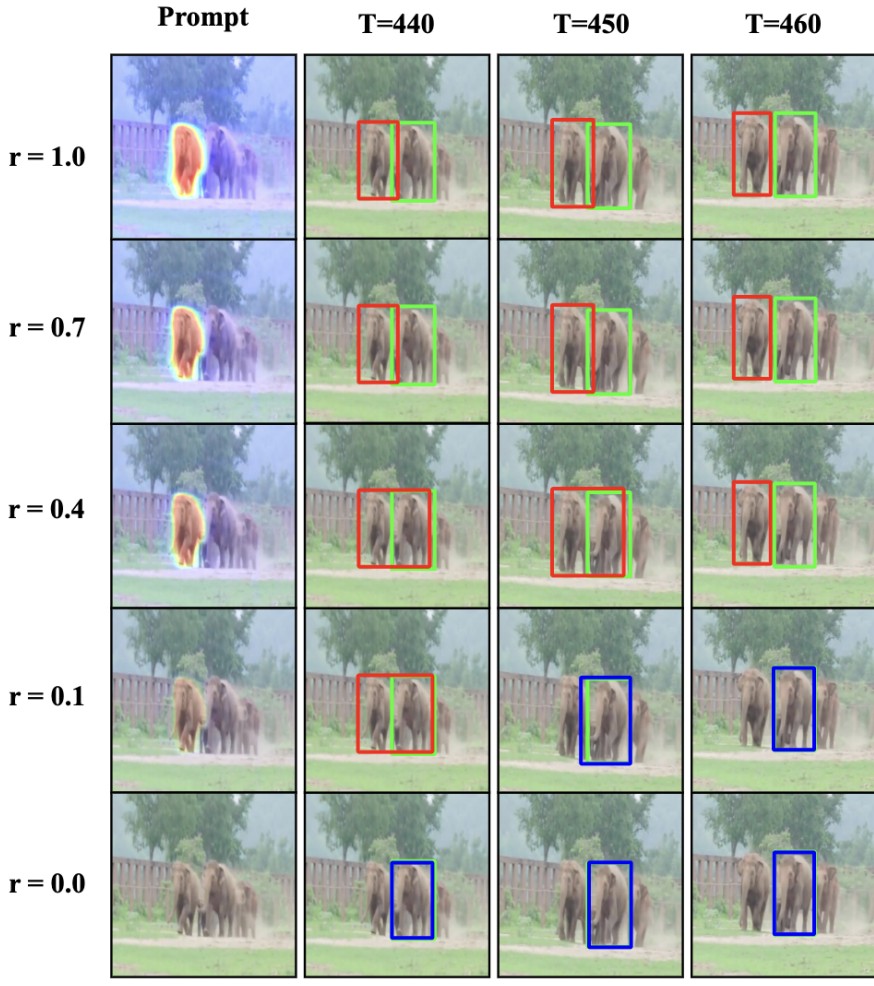

*Figure 6.* Qualitative failure analysis of continuous predictive fusion. Results are shown for different blending ratios $r$, where predictive prompts are continuously mixed with observation-driven cues. Green boxes indicate ground-truth targets, red boxes denote drifted predictions caused by accumulated predictive bias, and blue boxes represent correct tracking. Even with small $r$, erroneous predictive cues are gradually committed to memory, leading to delayed but inevitable failure.

## C. Qualitative Analysis of Reliability-Guided Predictive Memory

Figure 7 visualizes how RPM influences perception under varying observation quality. The heatmaps represent dense predictive prompts generated by RPM, which are selectively activated when observation-driven evidence degrades. When reliable visual cues are available, predictive prompts are suppressed, demonstrating that RPM uses predictive memory strictly as a conditional prior.

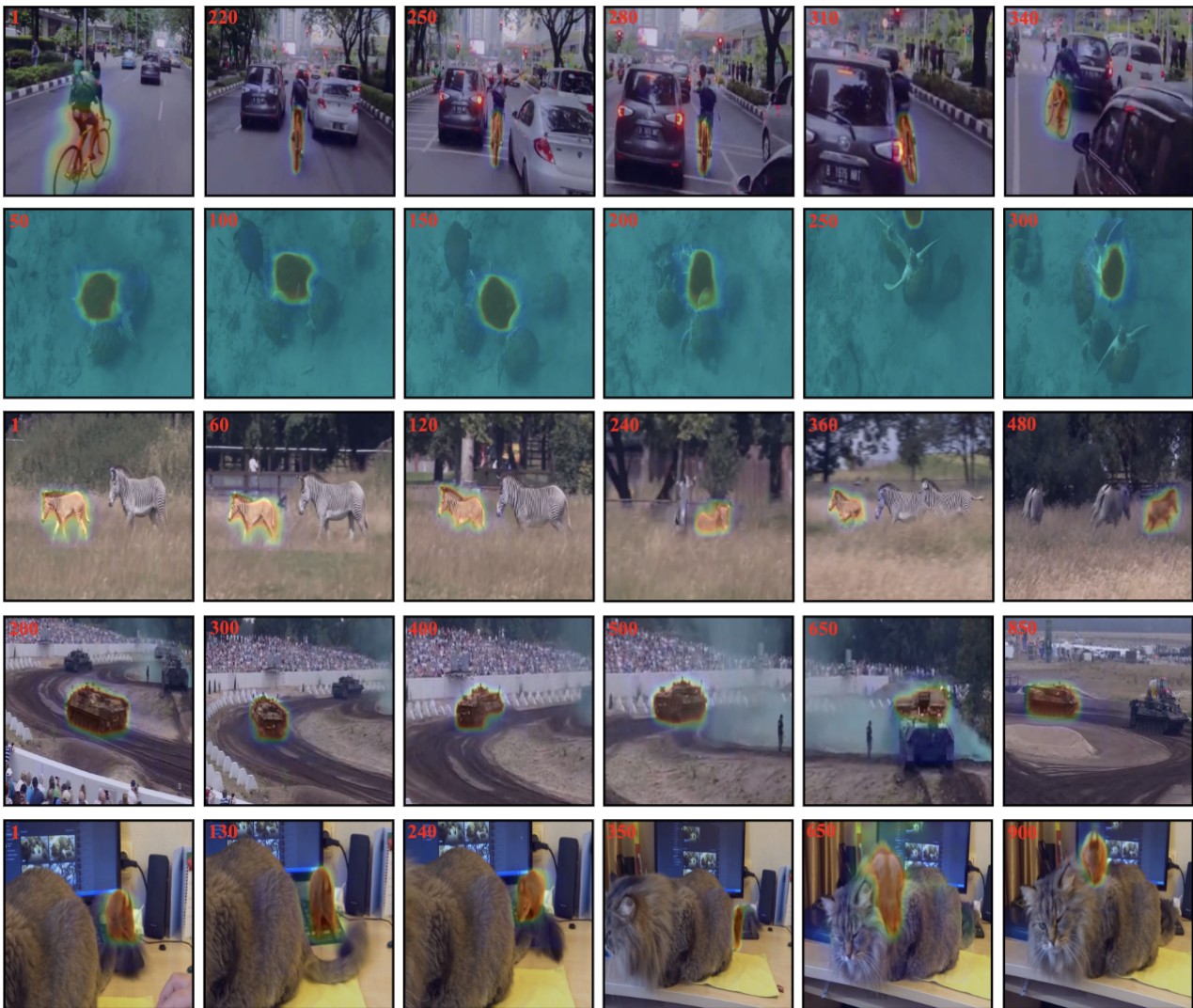

*Figure 7.* Visualization of Reliability-Guided Predictive Memory (RPM). Predictive prompts are activated only under occlusion or weak observation, guiding perception via reliable predictive priors. When observation quality is high, RPM suppresses predictive influence and relies entirely on observation-driven features.

