# OpenReview forum: "Memory as Dynamics: Learning Reliability-Guided Predictive Models for Online Video Perception"
_ICML.cc/2026/Conference — ICML 2026 regular_

### Official Review · Reviewer_koGd · 2026-03-01

**Soundness:** 3
**Presentation:** 2
**Significance:** 2
**Originality:** 2
**Overall Recommendation:** 4
**Confidence:** 3

**Summary:**

This paper addresses the drift problem commonly observed in tracking by proposing a reliability-guided mechanism for regulating memory usage. It introduces a gating strategy that integrates only reliable predictive cues into the memory system based on explicit reliability signals. By combining state-space predictive modeling with reliability-based gating, the authors present a plug-in RPM module that can be seamlessly integrated into existing architectures. The effectiveness of the approach is empirically validated through long-term tracking benchmarks and occlusion-aware analyses.

**Compliance With Llm Reviewing Policy:**

Affirmed.

**Final Justification:**

Although the methodological novelty is moderate and the paper lacks formal stability analysis, the approach is sound, clearly motivated, and empirically well validated, particularly under long-term occlusion. The rebuttal addressed my main concerns by clarifying design choices and strengthening the generality claims with additional experiments. While some theoretical aspects remain underdeveloped, the overall contribution is solid and practically meaningful. I therefore revise my recommendation to Weak Accept.

**Key Questions For Authors:**

(1) The RPM module is presented as a plug-in and general framework, yet all experiments are conducted on SAM2. Could the authors clarify whether RPM can be integrated into non-SAM memory-based video models, and whether they have preliminary evidence supporting its generality?

(2) Section 3.3 introduces the $z_t$​ prompt pathway via decoder injection, but the rationale for choosing decoder-level integration instead of direct memory-state updates is not fully explained. Could the authors clarify why decoder injection is structurally preferable and whether alternative integration strategies were considered?

(3) The paper argues that hard reliability gating mitigates error accumulation and drift, yet no formal analysis or long-term error characterization is provided. Under what assumptions does the proposed gating mechanism prevent unbounded drift, and how does it theoretically differ from continuous fusion in terms of error propagation?

(4) Training is performed on local temporal windows of length $K$, without explicitly modeling global time. Could the authors explain why locally trained predictive dynamics generalize to long-term robustness, particularly under extended occlusion?

**Limitations:**

Discussing scenarios in which the RPM module may fail to capture or handle certain cases, as well as clarifying whether these limitations are inherent to RPM itself or tied to the underlying SAM architecture, would provide deeper insight into the method and strengthen the discussion of its limitations.

**Strengths And Weaknesses:**

**Strengths**
- Clear and well-motivated problem formulation.
- An intuitive and structurally simple architecture.
- Reinterpreting predictive memory as a conditional prior governed by reliability rather than using prediction unconditionally.
- Designing reliability as a combination of complementary indicators rather than relying on a single confidence signal.
- Consistent performance improvements under long-term occlusion, full occlusion, and out-of-view conditions.
- Empirical validation through ablation that selective activation is superior to continuous fusion.

**Weaknesses**
- Despite the concrete empirical improvements, a substantial portion of the proposed methodology relies on existing models and design choices, which limits the novelty.
- Decoder confidence is used as a proxy for failure modes, but low confidence does not uniquely indicate occlusion or distractors, and their underlying causes are not explicitly modeled.
- Section 3.3 lacks sufficient explanation for why the $z_t$ prompt pathway is chosen, and why decoder injection is used instead of directly updating or overwriting the memory.
- Prediction errors in $z_t$ may originate from corrupted $o_{t−1}$ embedded in $s_{t−1}$, yet the reliability mechanism applies gating only to $z_t$ without explicitly accounting for potential contamination in $o_t$
- The core contribution is the RPM module, which is presented as a plug-in and general framework; however, evaluating it only on SAM limits the empirical support for its claimed generality.
- The absence of an explicit formulation of the overall system dynamics, particularly the core SAM memory update, makes it difficult to grasp the complete flow of the method.
- The parameters, including the thresholds $\tau$, have a heuristic nature and may require careful tuning.
- Training is conducted by sampling local time sequences of length $K$, without explicitly considering global time. Although the experiments clearly demonstrate strong long-term performance, it remains unclear why such global robustness emerges from locally trained dynamics.
- The paper does not provide any theoretical analysis or long-term error bounds for the predictive dynamics, despite claiming improved robustness over extended horizons.
- It remains unclear why hard gating theoretically prevents error accumulation or guarantees bounded drift.
- The conditions under which drift is provably suppressed are not specified, leaving the stability of the approach supported primarily by empirical evidence.

**Writing**
- The first paragraph of the Introduction addresses two distinct topics, namely background and the core problem; it is recommended to separate them into two paragraphs for clarity.
- The writing in the Introduction lacks a clear emphasis and presents claims without a well-defined information hierarchy, which reduces readability. It would be beneficial to foreground the central message and reorganize the section into a background–problem–contribution structure to establish a clearer logical flow.
- Occlusion Duration and Attribute-wise Analysis are better interpreted as detailed breakdowns of the main experimental results rather than as ablation studies.

---

> ### Author Rebuttal · Authors · 2026-03-31
>
> We thank the reviewer for the thoughtful feedback and for recognizing the clarity of our approach, and appreciate the questions clarifying the scope, limitations, and design choices.
>
> **On novelty (W1)**
>
> We agree that RPM builds on strong components such as SAM2 and will clarify this. The novelty lies in identifying reliability mismatch as a key failure mode and introducing a principle that separates short-term predictive influence from persistent memory, enabling recovery under degraded observations without corrupting memory.
>
> **On decoder confidence as a proxy (W2)**
>
> We agree that low decoder confidence does not uniquely identify specific causes such as occlusion or distractors. Our goal is not to classify failure causes, but to detect when observation quality becomes unreliable enough that predictive prompting should be treated conservatively.
> This abstraction is intentional: RPM is designed to handle degraded observations regardless of source, as reflected in consistent improvements (Table 6).
>
> **On the prompt pathway over direct memory overwrite (W3, Q2)**
>
> We considered alternatives such as direct memory updates or feature blending, but adopt decoder-level integration to ensure reversibility and contain prediction error. A decoder prompt affects only the current output, leaving memory intact, whereas direct updates risk error propagation. RPM thus keeps memory observation-driven and restricts predictive influence to decoding, as the predictive branch is inherently fallible under uncertainty.
>
> **On the prediction error from corrupted internal states (W4)**
>
> We agree that prediction errors may arise from corrupted states, and RPM does not eliminate this. Instead, it limits predictive influence when reliability is low. Since gating is applied only at the decoder prompt, errors affect only the current step while past memory remains intact. This is mitigated by reliability-guided anchoring, observation-driven signals, and sparse activation (3.71%, table 4), consistent with conservative use of the predictive branch.
>
> **On generality beyond SAM2 (W5, Q1)**
>
> We evaluate RPM in a non-SAM memory-based VOS model and supervised tracking.
> |Method|MOSE val|YTVOS2019 val|DAVIS2017 val|
> |-|-|-|-|
> |Cutie|68.3|86.5|88.8|
> |Cutie + RPM|**69.4**|**87.8**|**89.1**|
>
> |Method|LaSOT|LaSOText|GOT10k|
> |-|-|-|-|
> |OSTrack-384|71.1|50.5| 73.7|
> |OSTrack-384 + RPM|**72.3**|**52.1**|**76.4**|
>
> These results show that RPM extends beyond SAM2 as a general plug-in, while we accordingly limit our generality claim rather than overstate universality.
>
> **On system dynamics and presentation (W6, Writing)**
>
> We agree that the system flow is not sufficiently explicit. In the revision, we add a clearer high-level description of the full pipeline before the equations, restructure the Introduction into a clearer background, problem, contribution structure, and reclassify Occlusion Duration and Attribute-wise Analysis as detailed analyses.
>
> **On heuristic thresholds (W7)**
>
> We conduct threshold sensitivity analysis on SA-V val and LVOS v2 val (J&F).
> |$q_{\text{iou}}$|SA-V val|LVOS v2 val|
> |-|-|-|
> |0.3|79.3|83.8|
> |0.4|**79.5**|**83.9**|
> |0.5|**79.5**|83.7|
> |0.7|79.1|83.2|
>
> |$q_{\text{obj}}$|SA-V val|LVOS v2 val|
> |-|-|-|
> |0.0|79.2|83.7|
> |0.1|79.3|83.5|
> |0.2|79.2|83.6|
> |0.3|**79.5**|**83.9**|
>
> |$d_{\text{feature}}$|SA-V val|LVOS v2 val|
> |-|-|-|
> |0.2|79.4|83.7|
> |0.3|**79.5**|**83.9**|
> |0.4|79.3|83.5|
> |0.5|78.9|83.4|
>
> RPM shows stable performance over wide parameter ranges without requiring narrow threshold tuning.
>
> **On local training and global robustness (W8, Q4)**
>
> RPM learns a local decision rule for activating or suppressing predictive prompts based on current observation quality and consistency, rather than extrapolating long-term futures. Long-term robustness emerges from repeatedly applying this conservative rule, which defaults to suppression under uncertainty and limits drift accumulation. This is consistent with Table 3, where the gap between RPM and continuous fusion increases with occlusion duration.
>
> **On hard gating and long-horizon robustness (W9-W11, Q3)**
>
> We acknowledge that we do not provide a formal stability guarantee. Instead, we view hard gating as a conservative local mechanism that helps limit the influence of unreliable predictions in the recurrent update. In contrast to continuous fusion, where non-zero weights allow error propagation, hard gating can fully suppress predictive input.
>
> We further support this distinction through a controlled comparison with a lightweight MLP-based soft gating using the same reliability signals.
> |Method|AUC|
> |-|-|
> |Soft-Gating(Learned Scalar)|72.3|
> |Reliability-Guided Binary Gating(Ours)|**74.5**|
>
> **On failure cases and the source of limitations (Limitation)**
>
> RPM improves robustness but cannot eliminate errors under severe degradation, as reliability estimation in SAM is inherently difficult. Instead, it limits unreliable predictions rather than ensuring immunity to state corruption.

---

> > ### Author Rebuttal · Reviewer_koGd · 2026-04-01
> >
> > Although the methodological novelty is moderate and several components build upon existing architectures, and despite the absence of formal analysis on stability or long-term error behavior, the authors have clarified key design choices and strengthened the empirical validation in the rebuttal, including broader evaluations beyond a single backbone. While some aspects of the dynamics framing remain conceptually underdeveloped and certain design elements retain a heuristic nature, the core idea is clearly motivated, the architecture is intuitive, and the empirical improvements under long-term occlusion and degraded visibility are consistent and convincing. In light of these strengths and the authors’ clarifications, I revise my recommendation.

---

> > > ### Author Response · Authors · 2026-04-03
> > >
> > > We sincerely thank you for the positive reassessment and for acknowledging our clarifications and additional experiments.
> > > We will further improve the clarity of the dynamics formulation and explicitly discuss the heuristic aspects and limitations in the final version. Thank you again for your valuable feedback.

---

### Official Review · Reviewer_j6Ge · 2026-03-01

**Soundness:** 2
**Presentation:** 2
**Significance:** 3
**Originality:** 2
**Overall Recommendation:** 3
**Confidence:** 4

**Summary:**

This paper proposes an explicit modeling method for predictive dynamic reliability (RPM) to address common failure modes of predictive memory in online video perception, namely state contamination and identity drift after occlusion. The method combines a learned state space prediction module with a two-stage reliability gating system. The latter activates prediction cues only when observation confidence is low and prediction-observation feature consistency is high, supplemented by a reliability-guided spatial anchoring strategy. RPM is implemented based on SAM2.1 and tested with zero-shot visual object tracking on the LaSOT, LaSOText, and GOT-10k datasets. Results show that RPM significantly improves robustness, especially under occlusion conditions. Ablation experiments validate the effectiveness of each reliability component.

**Compliance With Llm Reviewing Policy:**

Affirmed.

**Final Justification:**

Thanks for the author's response. After considering the overall novelty of the paper and the rebuttal, I believe the current score is appropriate, so I will maintain it.

**Key Questions For Authors:**

Please pay attention to the Weakness section. If the author can resolve this and provide more evidence, I will consider raising the score.

**Limitations:**

No, please have the author add a discussion about limitations.

**Strengths And Weaknesses:**

Strengths:
1. Reinterprets memory in online video models as a dynamic latent process and introduces an explicit reliability-controlled interface between learned predictive dynamics and memory updates.

2. Combines state-space predictive modeling (Mamba) with a reliability gate based on decoder-derived IoU/objectness and a feature-consistency measure, and a spatial anchoring strategy that falls back to the last reliably observed frame.

3. Solid zero-shot evaluation on three standard tracking benchmarks with consistent improvements over strong SAM2-based baselines (SAMURAI, DAM4SAM, HiM2SAM).

Weakness:

Weaknesses:

1. Compared to other SAM-based methods, the paper's method introduces more parameters and additional threshold hyperparameters τ_iou, τ_obj, and τ_feature. While this provides a significant performance boost, it compromises generalization. The authors need to provide experiments on more scenarios to demonstrate this.

2. The formalization of the state-space model is rather abstract; the precise instantiation of formulas (4)–(6) using Mamba has a weak correlation with linear operators A, B, and C, and the feature consistency pruning/alignment process is not fully clear.

3. There is a lack of comparison with more VOT methods, such as MCITrack and ODTrackv2. Furthermore, the performance on VOT benchmarks is insufficient to fully demonstrate the method's advantages (performance and latency); a broader task set (e.g., VOS, MOT) would help verify the method's generality.

---

> ### Author Rebuttal · Authors · 2026-03-30
>
> We thank the reviewer for the careful reading and thoughtful comments. We particularly appreciate the recognition of our formulation and experimental strengths, as well as the clear identification of limitations.
>
> **On parameter increase and generalization (W1)**
>
> We agree that the additional parameters and thresholds should be justified. In RPM, the added capacity is limited to a lightweight predictive module, while the SAM2.1 backbone remains frozen, resulting in only a modest parameter increase without altering the core representation.
>
> The reliability thresholds are selected once on validation data and fixed across datasets and backbones, with stable performance across tested ranges (Table 7).
>
> To further examine generalization, we additionally evaluate the same threshold regime on VOS benchmarks using J&F.
> | $q_{\text{iou}}$ | SA-V | LVOS v2 val |
> |------------------|------|-------------|
> | 0.3              | 79.3 | 83.8        |
> | 0.4              | **79.5** | **83.9** |
> | 0.5              | **79.5** | 83.7     |
> | 0.7              | 79.1 | 83.2        |
>
> | $q_{\text{obj}}$ | SA-V val | LVOS v2 val |
> |------------------|----------|-------------|
> | 0.0              | 79.2     | 83.7        |
> | 0.1              | 79.3     | 83.5        |
> | 0.2              | 79.2     | 83.6        |
> | 0.3              | **79.5** | **83.9**    |
>
> | $d_{\text{feature}}$ | SA-V val | LVOS v2 val |
> |----------------------|----------|-------------|
> | 0.2                  | 79.4     | 83.7        |
> | 0.3                  | **79.5** | **83.9**    |
> | 0.4                  | 79.3     | 83.5        |
> | 0.5                  | 78.9     | 83.4        |
>
> Across broad parameter ranges, performance remains stable on VOS benchmarks without requiring narrow threshold tuning.
>
> **On the abstraction of the state-space formulation and Mamba2 instantiation (W2)**
>
> We thank the reviewer for pointing this out.
>
> In Eq. (4), $\bar{A}$ and $\bar{B}$ denote the discretized state transition and input. In Mamba2, the recurrence is selective: $\bar{B}$ is input-dependent, while $\bar{A}$ is modulated by the input-dependent step size $\Delta_{t,k}$.
> Specifically, $\bar{A} = \exp(\Delta_{t,k} \cdot A)$ models the discretized transition, and $\bar{C}$ projects the hidden state to the predictive latent. Eq. (6) performs temporal aggregation over $\{z_{t,k}\}$ via pooling. We will revise Section 3.2 to include an explicit correspondence table.
>
> Regarding alignment, we align $\tilde{z}_t$ and $o_t$ in a shared object-centric frame via target-centered cropping. The same crop is applied to both features, and their cosine distance is used as the prediction–observation consistency signal for gating.
>
> **On comparison with additional VOT methods and broader evaluation (W3)**
>
> We agree that comparisons with MCITrack and ODTrack are relevant and will add them to Table 1. At a comparable base-model scale (AUC / AO), we compare our method with MCITrack-B and ODTrack-B:
>
> | Tracker  | LaSOT | LaSOText | GOT-10k |
> |-------------|--------|-----------|------------|
> | MCITrack-B  | 75.3   | 54.6      | 77.9       |
> | ODTrack-B   | 73.2   | 52.4      | 77.0       |
> | Ours        | **74.5** | **60.9** | **79.7**   |
>
> These comparisons show competitive performance on GOT-10k and LaSOT, with the largest gains on the long-term LaSOText benchmark.
>
> We also analyze efficiency under a controlled SAM2.1-B setup on VOS benchmarks (J&F):
>
> | Dataset     | Activation Rate (%) | Latency (ms)     | Overhead (%) |
> |-------------|---------------------|------------------|--------------|
> | SA-V val    | 1.68                | 259.7 (+1.04)    | +0.40        |
> | LVOS v2 val | 1.92                | 263.1 (+1.18)    | +0.45        |
>
> On VOS benchmarks, predictive inference is activated only in a small fraction of frames, with minimal latency overhead, confirming sparse activation beyond tracking.
>
> To further assess generalization, we evaluate RPM on VOS benchmarks (J&F):
>
> | Model  | SA-V val | SA-V test | LVOS v2 val | LVOS v1 val | MOSE val | DAVIS2017 val | YTVOS2019 val |
> |--------|----------|-----------|-------------|-------------|----------|----------------|----------------|
> | SAM2.1 | 78.0     | 77.7      | 83.1        | 79.7        | 73.8     | 90.0           | 88.3           |
> | Ours   | **79.5** | **79.8**  | **83.9**    | **81.6**    | **74.6** | **90.7**       | **88.5**       |
>
> We additionally evaluate RPM on a supervised tracking baseline (AUC / AO):
>
> | Method              | LaSOT | LaSOText | GOT10k |
> |---------------------|-------|----------|--------|
> | OSTrack-384         | 71.1  | 50.5     | 73.7   |
> | OSTrack-384 + RPM   | **72.3** | **52.1** | **76.4** |
>
> These results show consistent gains across tracking and VOS, indicating that RPM generalizes beyond zero-shot SAM2 and benefits supervised tracking. Evaluating RPM in a meaningful MOT setting is beyond the present scope, as it involves additional components that make it difficult to isolate the contribution of RPM.

---

> > ### Author Rebuttal · Reviewer_j6Ge · 2026-04-03
> >
> > Thanks for the author's response. After considering the overall novelty of the paper and the rebuttal, I believe the current score is appropriate, so I will maintain it.

---

> > > ### Author Response · Authors · 2026-04-03
> > >
> > > We appreciate your response. However, we would kindly like to ask if you could point to specific aspects of our work that you found lacking in novelty, or any prior work that you believe covers our contributions.
> > > This would greatly help us understand your perspective and improve our work, regardless of the final outcome.

---

### Official Review · Reviewer_nFAb · 2026-03-06

**Soundness:** 3
**Presentation:** 1
**Significance:** 2
**Originality:** 3
**Overall Recommendation:** 4
**Confidence:** 3

**Summary:**

This paper studies predictive memory for online video perception and identifies a reliability mismatch where inaccurate predictions may contaminate memory under occlusion or uncertainty. It proposes Reliability-guided Predictive Memory (RPM), which models memory dynamics with a state-space predictor and regulates the use of predictions through a signal combining feature consistency and observation confidence.

**Compliance With Llm Reviewing Policy:**

Affirmed.

**Final Justification:**

Thanks for the authors’ response. The relevant parts should be clarified in the revised version. My concerns have been addressed.

**Key Questions For Authors:**

1. The method evaluates prediction reliability using observation-prediction consistency even when observations are already deemed unreliable. It is unclear whether this comparison can reliably estimate prediction correctness.

**Limitations:**

The predictive module appears to be designed for challenging cases such as occlusion. However, Tab. 2 only reports average latency. It remains unclear whether the proposed method can maintain real-time stability in corner cases with frequent occlusions where predictive inference may be repeatedly activated.

**Strengths And Weaknesses:**

Strengths:

1. The paper introduces a gating mechanism to regulate predictive memory and mitigate the impact of unreliable predictions.
2. The proposed design is relatively simple and can be incorporated into existing video perception frameworks without major architectural changes.


Weaknesses:

1. Presentation. The main text could benefit from clearer organization, particularly in the introduction. The current paragraph structure lacks clear focus and logical flow, making the presentation difficult to follow.
2. The reliability gating mechanism is heuristic-based. The framework relies on several fixed hyperparameters to regulate predictive prompt activation and spatial anchoring, which may limit adaptability and require manual tuning across different task scenarios.

---

> ### Author Rebuttal · Authors · 2026-03-29
>
> We thank the reviewer for the constructive feedback and for recognizing the simplicity and easy integration of our approach, and appreciate the insightful comments on presentation, reliability estimation, and limitations.
>
>
> **On the introduction (W1)**
>
> We agree that the introduction can be organized more clearly. In the revision, we restructure the introduction into three clearly separated parts:
>
> (a) background on memory in online video models and the limitations of static memory,
>
> (b) reliability mismatch as the central failure mode,
>
> (c) RPM as the proposed solution with concise contributions.
>
> Detailed related-work discussions move to Section 2 so that the narrative proceeds more directly from problem setting to failure mode to method.
>
> **On heuristic-based gating and threshold adaptability (W2)**
>
> We agree that the reliability gating mechanism is based on fixed thresholds, and that this raises a valid question about adaptability. The signals are derived from standard decoder outputs and prediction-observation consistency, and thus carry clear semantic meaning. The thresholds therefore operate on interpretable signals rather than arbitrary parameters.
>
> As shown in Table 7, thresholds fixed on LaSOT transfer to LaSOText and GOT-10k without retuning, despite differences in motion statistics and object categories.
>
> To further validate this behavior, we additionally analyze threshold sensitivity on VOS benchmarks (J&F).
>
>
> | $q_{\text{iou}}$ | SA-V val | LVOS v2 val |
> |-|-|-|
> | 0.3 | 79.3 | 83.8|
> | 0.4| **79.5** | **83.9** |
> | 0.5 | **79.5** | 83.7|
> | 0.7 | 79.1 | 83.2 |
>
>
> | $q_{\text{obj}}$ | SA-V val | LVOS v2 val |
> |------------------|----------|-------------|
> | 0.0              | 79.2     | 83.7        |
> | 0.1              | 79.3     | 83.5        |
> | 0.2              | 79.2     | 83.6        |
> | 0.3              | **79.5** | **83.9**    |
>
>
> | $d_{\text{feature}}$ | SA-V val | LVOS v2 val |
> |----------------------|----------|-------------|
> | 0.2                  | 79.4     | 83.7        |
> | 0.3                  | **79.5**     | **83.9**    |
> | 0.4                  | 79.3   | 83.5        |
> | 0.5                  | 78.9     | 83.4        |
>
>
> | $\alpha_{\text{crop}}$ | SA-V val | LVOS v2 val |
> |------------------------|----------|-------------|
> | 1.5| **79.5** | **84.0**    |
> | 2.0 | **79.5** | 83.9        |
> | 2.5 | 79.4     | 83.9        |
> | 3.0                    | 79.1     | 83.5        |
>
> Across broad ranges of parameters, performance varies only modestly, indicating that RPM operates in an empirically stable regime without requiring heavy per-task recalibration in practice rather than depending on narrow threshold tuning. We add an explicit limitations paragraph clarifying that recalibration may still be appropriate for tasks with substantially different observation statistics.
>
> **On reliability estimation under already-unreliable observations (Q1)**
>
> We thank the reviewer for raising this important point. A key aspect is the two-stage structure in Eq. (9): $d_{\text{feature}}$ is evaluated only when $q_{iou,t} < \tau_{iou}$, after the observation is identified as low-confidence. In this regime, $q_{\text{feature}}$ acts as a conservative non-divergence check that rejects clearly inconsistent cases, so the gate functions as a safety filter.
>
> This leads to different behaviors under common degradation scenarios:
>
> (1) Partial occlusion with well-aligned prediction: feature discrepancy stays low, making activation more likely to be appropriate.
>
> (2) Prediction drift: when the prediction deviates from the target, the discrepancy increases and the gate is suppressed.
>
> (3) Full occlusion: observations are often dominated by background or distractors, yielding low agreement with the predictive feature rather than spuriously activating the gate.
>
> We also consider the case where $s_{\text{t-1}}$ is already degraded, such that both prediction and observation are consistently incorrect. This is mitigated by the reliability-guided spatial anchoring in Section 3.5. When $q_{\text{obj}, t} < \tau_{\text{obj}}$, the crop anchor reverts to the most recently reliable frame, thereby breaking the drift–crop feedback loop. This effect is reflected in Table 9 (previous-frame anchoring: AUC 73.8; reliability-guided anchoring: 74.5).
>
> **On stability under frequent occlusion (Limitation)**
>
> We agree that average latency alone does not fully characterize behavior under repeated occlusion. In RPM, however, predictive memory is activated only when observation reliability degrades rather than at every frame. As shown in Table 8, the predictive gate is activated in only 3.71% of LaSOT frames, which is consistent with the intended sparse and conservative use of the predictive branch. Accordingly, the average runtime overhead remains small (+0.82% over SAM2.1-B in Table 2). We will clarify in the revision that this is an empirical efficiency observation rather than a formal guarantee under all occlusion patterns.

---

> > ### Author Rebuttal · Reviewer_nFAb · 2026-04-04
> >
> > Thanks for the response. After re-reading Section 3.5, I find that the clarification provided in the rebuttal to Q1 is not explicitly stated in the paper, which may be misleading to readers. If this paper is eventually accepted, I hope this point can be clarified more concretely in the revised version.

---

> > > ### Author Response · Authors · 2026-04-04
> > >
> > > Thank you for re-reading the paper carefully and for this constructive suggestion. If accepted, we will revise Section 3.5 to explicitly clarify the role of the reliability signal and its interpretation, ensuring it is not misleading to readers. We believe your suggestions have greatly improved the quality of our paper, and hope that our responses have sufficiently addressed your concerns.

---

### Official Review · Reviewer_Dwpr · 2026-03-13

**Soundness:** 4
**Presentation:** 3
**Significance:** 4
**Originality:** 3
**Overall Recommendation:** 5
**Confidence:** 4

**Summary:**

This paper presents Memory as Dynamics: Learning Reliability-Guided Predictive Models for Online Video Perception, where the authors propose a framework for improving robustness in online video perception models. The authors observe that the recent predictive memory approaches often inject predicted features into memory without considering their reliability, which can lead to memory contamination and identity drift, especially during occlusions or sudden motion. To address this, the paper formulates online video perception as a sequential latent state estimation problem and introduces a reliability-aware mechanism that regulates when predictive memory should influence the model. The framework is implemented on top of a frozen SAM2 backbone. A state-space predictive dynamics module generates a latent prediction of future memory states, which is converted into decoder-aligned prompts. These predictive prompts are injected into the segmentation decoder only when a reliability condition is satisfied, based on observation confidence, prediction-observation feature consistency, and object confidence. The method is evaluated on multiple object tracking benchmarks, and shows improvements over several SAM2-based baselines.

**Compliance With Llm Reviewing Policy:**

Affirmed.

**Final Justification:**

The authors have provided a solid response that addresses my concerns, and the paper is now in a much improved state.

**Key Questions For Authors:**

Have the authors explored the possibility of learning the reliability function end-to-end instead of using manually defined thresholds?

The paper argues that continuous or soft fusion of predictive and observation features can lead to error accumulation. Could the authors provide a quantitative comparison of soft gating and proposed threshold-based activation mechanism?

**Limitations:**

Yes

**Strengths And Weaknesses:**

***Strengths***:

- The paper identifies a practical limitation in predictive memory systems for video perception: predictive signals are often injected into memory without assessing their reliability. The paper frames this issue as a reliability mismatch between predictive dynamics and observation-driven updates, which is intuitive and well motivated.
- The proposed reliability-guided activation mechanism provides a simple yet effective strategy to control when predictive information should influence the model’s prediction. By gating predictive prompts based on observation confidence and prediction-observation consistency, the framework introduces a practical way to address error accumulation in memory-based video models.
- Experiments on challenging tracking benchmarks like LaSOT, LaSOText, and GOT-10k show consistent improvements over several SAM2-based tracking baselines. The gains are particularly noticeable under occlusion scenarios, which aligns well with the motivation.

***Weaknesses:***

- Reliability estimation relies on heuristic thresholds. The reliability signal is computed using manually defined thresholds based on IoU confidence, object confidence, and feature similarity. These heuristic choices are not learned end-to-end and may require tuning for different datasets or tasks.
- Some notations are undefined. For example, the symbols A and B in Equation (4) are not defined; they may correspond to the Mamba formulation, but this is not explicitly stated, which makes their meaning unclear.
- Limited empirical comparison with alternative fusion strategies: The paper argues that continuous or soft fusion of predictive features leads to error accumulation, but this claim is mainly supported through qualitative discussion. A quantitative comparison with soft gating would help substantiate this design choice.
- The paper presents RPM as a general framework for online video perception, yet empirical evaluation is limited to visual object tracking. While tracking is indeed a representative setting for memory-based video models, the claim of general applicability would be stronger if validated on additional tasks like video segmentation.

---

> ### Author Rebuttal · Authors · 2026-03-29
>
> We thank the reviewer for the detailed assessment and for recognizing the motivation and effectiveness of our approach, and appreciate the insightful comments on reliability estimation, fusion strategies, generality, and presentation.
>
> **On heuristic thresholds and end-to-end learning (W1, Q1)**
>
> We agree that the reliability function is deterministic rather than end-to-end learned, as a deliberate design choice. This is motivated by two factors. First, the signals are directly derived from the decoder or prediction–observation consistency, providing clear semantic meaning rather than acting as free parameters. Second, Table 7 shows AUC variation below 0.5, and thresholds fixed on LaSOT transfer directly across datasets, indicating stable operation without per-dataset tuning.
>
> To further examine cross-task threshold stability, we additionally evaluate sensitivity on VOS benchmarks (J&F).
>
> | $q_{\mathrm{iou}}$ | SA-V val | LVOS v2 val |
> |--|--|--|
> | 0.3 | 79.3 | 83.8 |
> | 0.4 | **79.5** | **83.9** |
> | 0.5 | **79.5** | 83.7 |
> | 0.7 | 79.1 | 83.2 |
>
> | $q_{\mathrm{obj}}$ | SA-V val | LVOS v2 val |
> |--|-|--|
> | 0.0 | 79.2 | 83.7 |
> | 0.1 | 79.3 | 83.5 |
> | 0.2 | 79.2 | 83.6 |
> | 0.3 | **79.5** | **83.9** |
>
> | $d_{\mathrm{feature}}$ | SA-V val | LVOS v2 val |
> |----|--|-|
> | 0.2 | 79.4 | 83.7 |
> | 0.3 | **79.5** | **83.9** |
> | 0.4 | 79.3 | 83.5 |
> | 0.5 | 78.9 | 83.4 |
>
> Across broad ranges of reliability signals, performance varies only marginally: fixing any value within the tested range yields similar performance across datasets, indicating that RPM does not depend on per-dataset tuning.
>
> The distinction from a learned gate lies in the nature of the decision. A learned gate must infer suppression from sparse gradient signals in the few frames where it should close. In contrast, the binary gate directly applies reliability signals at inference time, defining suppression based on their semantic meaning rather than learning it indirectly. This design aligns with the sparse activation regime observed in practice (~4% in Table 8), where suppression events are rare.
>
> **On undefined notation (W2)**
>
> We thank the reviewer for pointing this out. In Eq. (4), $\bar{A}$ and $\bar{B}$ denote the discretized state-transition and input matrices. In the Mamba2 instantiation, the recurrence is selective: $\bar{B}$ is input-dependent, while $\bar{A}$ is modulated through the input-dependent discretization step $\Delta_{t,k}$. In the revision, we clarify this in Section 3.2 and add a correspondence table linking Eq. (4) to the Mamba2 parameterization (Appendix A.3).
>
> **On quantitative comparison with soft gating (W3, Q2)**
>
> We compare binary activation with a learned scalar soft gate under the same LaSOT setup and frozen-backbone protocol, keeping all other RPM components fixed. The soft gate takes the same three reliability signals as input and is implemented as a lightweight MLP with sigmoid output, isolating the effect of the gating rule.
> | Method | AUC |
> |--------|-----|
> | SAM2.1-B | 66.0 |
> | Always-on Prediction Injection | 70.5 |
> | Soft-Gating (Learned Scalar) | 72.3 |
> | Reliability-Guided Binary Gating (Ours) | **74.5** |
>
> These results show that soft interpolation learns some suppression behavior, but remains less effective than hard suppression in a recursive memory setting, with binary gating outperforming soft gating. Any non-zero interpolation weight still allows predictive error to influence future states, whereas binary suppression turns predictive influence off entirely when reliability is low.
>
> This is further supported by the learned α distribution:
> | Method | $\alpha < 0.2$ | $0.2 \le \alpha < 0.5$ | $0.5 \le \alpha < 0.8$ | $\alpha \ge 0.8$ |
> |--------|---------------|----------------------|----------------------|----------------|
> | Continuous Fusion | 5.6% | 32.6% | 45.1% | 16.7% |
>
> Only 5.6% of frames fall into α<0.2, while most remain in the intermediate range (0.2≤α<0.8). This suggests that learned gating only attenuates unreliable predictions, allowing greater residual error propagation under recursive memory updates than binary suppression. This is also consistent with the qualitative failure analysis in Appendix B.5.
>
> **On generalization beyond tracking (W4)**
>
> We agree that the original quantitative evaluation is centered on tracking. To further assess generalization, we evaluate it on standard VOS benchmarks without task-specific fine-tuning.
> | Model | SA-V val | SA-V test | LVOS v2 val | LVOS v1 val | MOSE val | DAVIS2017 val | YTVOS2019 val |
> |-------|----------|-----------|-------------|-------------|----------|---------------|----------------|
> | SAM2.1 | 78.0 | 77.7 | 83.1 | 79.7 | 73.8 | 90.0 | 88.3 |
> | Ours | **79.5** | **79.8** | **83.9** | **81.6** | **74.6** | **90.7** | **88.5** |
>
> Compared with SAM2.1, RPM consistently improves J&F across all benchmarks, providing direct additional evidence that the reliability-guided mechanism transfers beyond tracking without task-specific redesign.

---

> > ### Author Rebuttal · Reviewer_Dwpr · 2026-04-05
> >
> > Fully resolved. I'll raise the score.

---

> > > ### Author Response · Authors · 2026-04-05
> > >
> > > Thank you very much for your positive feedback and for taking the time to reconsider our paper. We sincerely appreciate your thoughtful evaluation and your decision to raise the score.

---

### Decision · Program_Chairs · 2026-04-30

**Decision:**

Accept (regular)

**Comment:**

Overall the reviewers are positive (5, 4, 3, 4) about this paper for identifying and addressing the critical issue of identity drift in online video perception through an intuitive gating mechanism. While the methodological novelty is moderate and the paper lacks formal stability analysis, the approach is sound, clearly motivated, and empirically well validated, particularly under long-term occlusion. The rebuttal successfully clarified key design choices and demonstrated the framework's stability and generality across diverse backbones and tasks. This meta-reviewer agrees that the overall contribution is solid and practically meaningful, as the consistent empirical improvements outweigh concerns regarding the heuristic nature of the design.